# ADAPTIVE FEDERATED LEARNING WITH AUTO-TUNED CLIENTS

**Junhyung Lyle Kim**[*] **Mohammad Taha Toghani**[†] **César A. Uribe**[†] **& Anastasios Kyrillidis**[*]
[*]Department of Computer Science, [†]Department of Electrical and Computer Engineering
Rice University, Houston, TX 77005, USA
{jlylekim, mttoghani, cauribe, anastasios}@rice.edu

## ABSTRACT

Federated learning (FL) is a distributed machine learning framework where the global model of a central server is trained via multiple collaborative steps by participating clients without sharing their data. While being a flexible framework, where the distribution of local data, participation rate, and computing power of each client can greatly vary, such flexibility gives rise to many new challenges, especially in the hyperparameter tuning on the client side. We propose $\Delta$-SGD, a simple step size rule for SGD that enables each client to use its own step size by adapting to the local smoothness of the function each client is optimizing. We provide theoretical and empirical results where the benefit of the client adaptivity is shown in various FL scenarios.

## 1 INTRODUCTION

Federated Learning (FL) is a machine learning framework that enables multiple clients to collaboratively learn a global model in a distributed manner. Each client trains the model on their local data, and then sends only the updated model parameters to a central server for aggregation. Mathematically, FL aims to solve the following optimization problem:

$$\min_{x \in \mathbb{R}^d} f(x) := \frac{1}{m} \sum_{i=1}^{m} f_i(x), \tag{1}$$

where $f_i(x) := \mathbb{E}_{z \sim \mathcal{D}_i}[F_i(x, z)]$ is the loss function of the $i$-th client, and $m$ is the number of clients.

A key property of FL its flexibility in how various clients participate in the overall training procedure. The number of clients, their participation rates, and computing power available to each client can vary and change at any time during the training. Additionally, the local data of each client is not shared with others, resulting in better data privacy (McMahan et al., 2017; Agarwal et al., 2018).

While advantageous, such flexibility also introduces a plethora of new challenges, notably: $i$) how the server aggregates the local information coming from each client, and $ii$) how to make sure each client meaningfully "learns" using their local data and computing device. The first challenge was partially addressed in Reddi et al. (2021), where adaptive optimization methods such as Adam (Kingma & Ba, 2014) was utilized in the aggregation step. Yet, the second challenge remains largely unaddressed.

Local data of each client is not shared, which intrinsically introduces heterogeneity in terms of the size and the distribution of local datasets. That is, $\mathcal{D}_i$ differs for each client $i$, as well as the number of samples $z \sim \mathcal{D}_i$. Consequently, $f_i(x)$ can vastly differ from client to client, making the problem in (1) hard to optimize. Moreover, the sheer amount of local updates is far larger than the number of aggregation steps, due to the high communication cost—typically 3-4$\times$ orders of magnitude more expensive than local computation—in distributed settings (Lan et al., 2020).

As a result, extensive fine-tuning of the client optimizers is often required to achieve good performance in FL scenarios. For instance, experimental results of the well-known `FedAvg` algorithm were obtained after performing a grid-search of typically 11-13 step sizes of the clients' SGD (McMahan et al., 2017, Section 3), as SGD (and its variants) are highly sensitive to the step size (Toulis & Airoldi, 2017; Assran & Rabbat, 2020; Kim et al., 2022b). Similarly, in Reddi et al. (2021), 6 different client

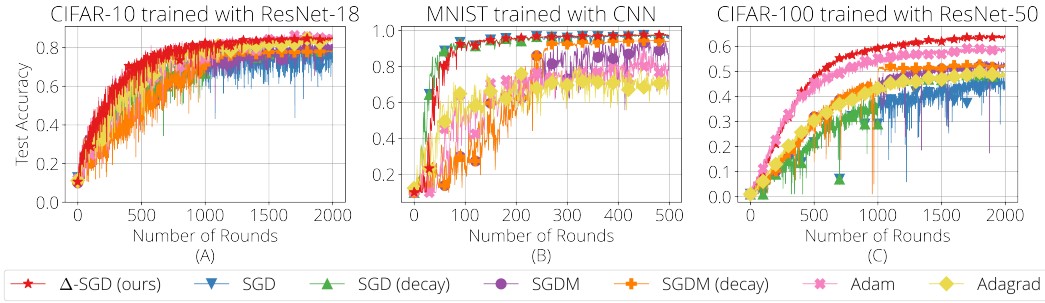

Figure 1: *Illustration of the effect of not properly tuning the client step sizes.* In (A), each client optimizer uses the best step size from grid-search. Then, the same step size from (A) is intentionally used in settings (B) and (C). Only Δ-SGD works well across all settings without additional tuning.

step sizes were grid-searched for different tasks, and not surprisingly, each task requires a different client step size to obtain the best result, regardless of the server-side adaptivity (Reddi et al., 2021, Table 8). Importantly, even these "fine-tunings" are done under the setting that *all clients use the same step size*, which is sub-optimal, given that $f_i$ can be vastly different per client; we analyze this further in Section 2.

**Initial examination as motivation.** The implication of not properly tuning the client optimizer is highlighted in Figure 1. We plot the progress of test accuracies for different client optimizers, where, for all the other test cases, we intentionally use *the same step size rules that were fine-tuned for the task in Figure 1(A)*. There, we train a ResNet-18 for CIFAR-10 dataset classification within an FL setting, where the best step sizes for each (client) optimizer was used after grid-search; we defer the experimental details to Section 4. Hence, all methods perform reasonably well, although Δ-SGD, our proposed method, achieves noticeably better test accuracy when compared to non-adaptive SGD variants –e.g., a 5% difference in final classification accuracy from SGDM– and comparable final accuracy only with adaptive SGD variants, like Adam and Adagrad.

In Figure 1(B), we train a shallow CNN for MNIST classification, using the *same* step sizes from (A). MNIST classification is now considered an "easy" task, and therefore SGD with the same constant and decaying step sizes from Figure 1(A) works well. However, with momentum, SGDM exhibits highly oscillating behavior, which results in slow progress and poor final accuracy, especially without decaying the step size. Adaptive optimizers, like Adam and Adagrad, show similar behavior, falling short in achieving good final accuracy, compared to their performance in the case of Figure 1(A).

Similarly, in Figure 1(C), we plot the test accuracy progress for CIFAR-100 classification trained on a ResNet-50, again using the same step size rules as before. Contrary to Figure 1(B), SGD with momentum (SGDM) works better than SGD, both with the constant and the decaying step sizes. Adam becomes a "good optimizer" again, but its "sibling," Adagrad, performs worse than SGDM. On the contrary, our proposed method, Δ-SGD, which we introduce in Section 3, achieves superior performance in all cases without any additional tuning.

The above empirical observations beg answers to important and non-trivial questions in training FL tasks using variants of SGD methods as the client optimizer: *Should the momentum be used? Should the step size be decayed? If so, when?* Unfortunately, Figure 1 indicates that the answers to these questions highly vary depending on the setting; once the dataset itself or how the dataset is distributed among different clients changes, or once the model architecture changes, the client optimizers have to be properly re-tuned to ensure good performance. Perhaps surprisingly, the same holds for adaptive methods like Adagrad (Duchi et al., 2011) and Adam (Kingma & Ba, 2014).

**Our hypothesis and contributions.** Our paper takes a stab in this direction: we propose DELTA-SGD (**D**istribut**E**d **L**ocali**T**y **A**daptive SGD), a simple adaptive distributed SGD scheme, that can automatically tune its step size based on the available local data. We will refer to our algorithm as Δ-SGD in the rest of the text. Our contributions can be summarized as follows:

• We propose Δ-SGD, which has two implications in FL settings: $i$) each client can use its own step size, and $ii$) each client's step size *adapts to the local smoothness of $f_i$* –hence LocaliTy Adaptive– and can even *increase* during local iterations. Moreover, due to the simplicity of the proposed step

size, $\Delta$-SGD is *agnostic to the loss function and the server optimizer*, and thus can be combined with methods that use different loss functions such as FedProx (Li et al., 2020) or MOON (Li et al., 2021), or adaptive server methods such as FedAdam (Reddi et al., 2021).

- We provide convergence analysis of $\Delta$-SGD in a general nonconvex setting (Theorem 1). We also prove convergence in convex setting; due to space constraints, we only state the result of the general nonconvex case in Theorem 1, and defer other theorems and proofs to Appendix A.

- We evaluate our approach on several benchmark datasets and demonstrate that $\Delta$-SGD achieves superior performance compared to other state-of-the-art FL methods. Our experiments show that $\Delta$-SGD can effectively adapt the client step size to the underlying local data distribution, and achieve convergence of the global model *without any additional tuning*. Our approach can help overcome the client step size tuning challenge in FL and enable more efficient and effective collaborative learning in distributed systems.

## 2 PRELIMINARIES AND RELATED WORK

**A bit of background on optimization theory.** Arguably the most fundamental optimization algorithm for minimizing a function $f(x) : \mathbb{R}^d \to \mathbb{R}$ is the gradient descent (GD), which iterates with the step size $\eta_t$ as: $x_{t+1} = x_t - \eta_t \nabla f(x_t)$. Under the assumption that $f$ is *globally $L$-smooth*, that is:

$$\|\nabla f(x) - \nabla f(y)\| \leqslant L \cdot \|x - y\| \quad \forall x, y \in \mathbb{R}^d, \tag{2}$$

the step size $\eta_t = \frac{1}{L}$ is the "optimal" step size for GD, which converges for convex $f$ at the rate:

$$f(x_{t+1}) - f(x^\star) \leqslant \frac{L\|x_0 - x^\star\|^2}{2(2t + 1)}. \tag{3}$$

**Limitations of popular step sizes.** In practice, however, the constants like $L$ are rarely known (Boyd et al., 2004). As such, there has been a plethora of efforts in the optimization community to develop a universal step size rule or method that does not require a priori knowledge of problem constants so that an optimization algorithm can work as "plug-and-play" (Nesterov, 2015; Orabona & Pál, 2016; Levy et al., 2018). A major lines of work on this direction include: $i)$ line-search (Armijo, 1966; Paquette & Scheinberg, 2020), $ii)$ adaptive learning rate (e.g., Adagrad (Duchi et al., 2011) and Adam (Kingma & Ba, 2014)), and $iii)$ Polyak step size (Polyak, 1969), to name a few.

Unfortunately, the pursuit of finding the "ideal" step size is still active (Loizou et al., 2021; Defazio & Mishchenko, 2023; Bernstein et al., 2023), as the aforementioned approaches have limitations. For instance, to use line search, solving a sub-routine is required, inevitably incurring additional evaluations of the function and/or the gradient. To use methods like Adagrad and Adam, the knowledge of $D$, the distance from the initial point to the solution set, is required to ensure good performance (Defazio & Mishchenko, 2023). Similarly, Polyak step size requires knowledge of $f(x^\star)$ (Hazan & Kakade, 2019), which is not always known a priori, and is unclear what value to use for an arbitrary loss function and model; see also Section 4 where not properly estimating $D$ or $f(x^\star)$ resulting in suboptimal performance of relevant client optimizers in FL scenarios.

**Implications in distributed/FL scenarios.** The global $L$-smoothness in (2) needs to be satisfied for all $x$ and $y$, implying even finite $L$ can be arbitrarily large. This results in a small step size, leading to slow convergence, as seen in (3). Malitsky & Mishchenko (2020) attempted to resolve this issue by proposing a step size for (centralized) GD that depends on the *local smoothness* of $f$, which by definition is smaller than the global smoothness constant $L$ in (2). $\Delta$-SGD is inspired from this work.

Naturally, the challenge is aggravated in the distributed case. To solve (1) under the assumption that each $f_i$ is $L_i$-smooth, the step size of the form $1/L_{\max}$ where $L_{\max} := \max_i L_i$ is often used (Yuan et al., 2016; Scaman et al., 2017; Qu & Li, 2019), to ensure convergence (Uribe et al., 2020). Yet, one can easily imagine a situation where $L_i \gg L_j$ for $i \neq j$, in which case the convergence of $f_j(x)$ can be arbitrarily slow by using step size of the form $1/L_{\max}$.

Therefore, to ensure that each client learns useful information in FL settings as in (1), ideally: $i)$ each agent should be able to use its own step size, instead of crude ones like $1/L_{\max}$ for all agents; $ii)$ the individual step size should be "locally adaptive" to the function $f_i$, (i.e., even $1/L_i$ can be too crude, analogously to the centralized GD); and $iii)$ the step size should not depend on problem constants

like $L_i$ or $\mu_i$, which are often unknown. In Section 3, we introduce **D**istribut**E**d **L**ocali**T**y **A**daptive SGD ($\Delta$-SGD), which satisfies all of the aforementioned desiderata.

**Related work on FL.** The FL literature is vast; here, we focus on the results that closely relate to our work. The FedAvg algorithm (McMahan et al., 2017) is one of the simplest FL algorithms, which averages client parameters after some local iterations. Reddi et al. (2021) showed that FedAvg is a special case of a meta-algorithm, where both the clients and the server use SGD optimizer, with the server learning rate being 1. To handle data heterogeneity and model drift, they proposed using adaptive optimizers *at the server*. This results in algorithms such as FedAdam and FedYogi, which respectively use the Adam (Kingma & Ba, 2014) and Yogi (Zaheer et al., 2018) as server optimizers. Our approach is orthogonal *and* complimentary to this, as we propose a *client adaptive optimizer* that can be easily combined with these server-side aggregation methods (c.f., Appendix B.4).

Other approaches have handled the heterogeneity by changing the loss function. FedProx (Li et al., 2020) adds an $\ell_2$-norm proximal term to to handle data heterogeneity. Similarly, MOON (Li et al., 2021) uses the model-contrastive loss between the current and previous models. Again, our proposed method can be seamlessly combined with these approaches (c.f., Table 2b, Appendix B.6 and B.5).

The closest related work to ours is a concurrent work by Mukherjee et al. (2023). There, authors utilize the Stochastic Polyak Stepsize (Loizou et al., 2021) in FL settings. We do include SPS results in Section 4. There are some previous works that considered client adaptivity, including AdaAlter (Xie et al., 2019a) and Wang et al. (2021). Both works utilize an adaptive client optimizer similar to Adagrad. However, AdaAlter incurs twice the memory compared to FedAvg, as AdaAlter requires communicating both the model parameters as well as the "client accumulators"; similarly, Wang et al. (2021) requires complicated local *and* global correction steps to achieve good performance. In short, previous works that utilize client adaptivity require modification in server-side aggregation; without such heuristics, Adagrad (Duchi et al., 2011) exhibits suboptimal performance, as seen in Table 1.

Lastly, another line of works attempts to handle heterogeneity by utilizing control-variates (Praneeth Karimireddy et al., 2019; Liang et al., 2019; Karimireddy et al., 2020), a similar idea to variance reduction from single-objective optimization (Johnson & Zhang, 2013; Gower et al., 2020). While theoretically appealing, these methods either require periodic full gradient computation or are not usable under partial client participation, as all the clients must maintain a state throughout all rounds.

## 3 DELTA($\Delta$)-SGD: DISTRIBUTED LOCALITY ADAPTIVE SGD

We now introduce $\Delta$-SGD. In its simplest form, client $i$ at communication round $t$ uses the step size:

$$\eta_t^i = \min\left\{ \frac{\|x_t^i - x_{t-1}^i\|}{2\|\nabla f_i(x_t^i) - \nabla f_i(x_{t-1}^i)\|}, \sqrt{1 + \theta_{t-1}^i}\, \eta_{t-1}^i \right\}, \quad \theta_{t-1}^i = \eta_{t-1}^i / \eta_{t-2}^i. \tag{4}$$

The first part of $\min\{\cdot, \cdot\}$ approximates the (inverse of) local smoothness of $f_i$,[1] and the second part controls how fast $\eta_t^i$ can increase. Indeed, $\Delta$-SGD with (4) enjoys the following decrease in the Lyapunov function:

$$\|x_{t+1} - x^\star\|^2 + \frac{1}{2m}\sum_{i=1}^m \|x_{t+1}^i - x_t^i\|^2 + \frac{2}{m}\sum_{i=1}^m \left[\eta_{t+1}^i \theta_{t+1}^i \left(f_i(x_t^i) - f_i(x^\star)\right)\right]$$

$$\leqslant \|x_t - x^\star\|^2 + \frac{1}{2m}\sum_{i=1}^m \|x_t^i - x_{t-1}^i\|^2 + \frac{2}{m}\sum_{i=1}^m \left[\eta_t^i \theta_t^i \left(f_i(x_{t-1}^i) - f_i(x^\star)\right)\right], \tag{5}$$

when $f_i$'s are assumed to be convex (c.f., Theorem 5 in Appendix A). For the FL settings, we extend (4) by including stochasticity and local iterations, as summarized in Algorithm 1 and visualized in Figure 7. For brevity, we use $\tilde{\nabla} f_i(x) = \frac{1}{|\mathcal{B}|}\sum_{z\in\mathcal{B}} \nabla F_i(x, z)$ to denote the stochastic gradients with batch size $|\mathcal{B}| = b$.

---

[1]Notice that $\|\nabla f_i(x_t^i) - \nabla f_i(x_{t-1}^i)\| \leqslant \frac{1}{2\eta_t^i}\|x_t^i - x_{t-1}^i\| \approx \tilde{L}_{i,t}\|x_t^i - x_{t-1}^i\|$.

---

**Algorithm 1** DELTA($\Delta$)-SGD: **D**istribut**E**d **L**ocali**T**y **A**daptive SGD

---

1: **input**: $x_0 \in \mathbb{R}^d$, $\eta_0, \theta_0, \gamma > 0$, and $p \in (0, 1)$.
2: **for** each round $t = 0, 1, \ldots, T-1$ **do**
3:     sample a subset $\mathcal{S}_t$ of clients with size $|\mathcal{S}_t| = p \cdot m$
4:     **for** each machine in parallel for $i \in \mathcal{S}_t$ **do**
5:         set $x_{t,0}^i = x_t$
6:         set $\eta_{t,0}^i = \eta_0$   and   $\theta_{t,0}^i = \theta_0$
7:         **for** local step $k \in [K]$ **do**
8:             $x_{t,k}^i = x_{t,k-1}^i - \eta_{t,k-1}^i \tilde{\nabla} f_i(x_{t,k-1}^i)$                $\triangleright$ update local parameter with $\Delta$-SGD
9:             $\eta_{t,k}^i = \min\left\{ \frac{\gamma \|x_{t,k}^i - x_{t,k-1}^i\|}{2\|\tilde{\nabla} f_i(x_{t,k}^i) - \tilde{\nabla} f_i(x_{t,k-1}^i)\|}, \sqrt{1 + \theta_{t,k-1}^i}\, \eta_{t,k-1}^i \right\}$
10:             $\theta_{t,k}^i = \eta_{t,k}^i / \eta_{t,k-1}^i$           $\triangleright$ (line 9 & 10) update locality adaptive step size
11:         **end for**
12:     **end for**
13:     $x_{t+1} = \frac{1}{|\mathcal{S}_t|} \sum_{i \in \mathcal{S}_t} x_{t,K}^i$                         $\triangleright$ server-side aggregation
14: **end for**
15: **return** $x_T$

---

We make a few remarks of Algorithm 1. First, the input $\theta_0 > 0$ can be quite arbitrary, as it can be corrected, per client level, in the first local iteration (line 10); similarly for $\eta_0 > 0$, although $\eta_0$ should be sufficiently small to prevent divergence in the first local step. Second, we include the "amplifier" $\gamma$ to the first condition of step size (line 9), but this is only needed for Theorem 1.[2] Last, $\tilde{\nabla} f_i(x_{t,k-1}^i)$ shows up twice: in updating $x_{t,k}^i$ (line 8) and $\eta_{t,k}^i$ (line 9). Thus, one can use the same or different batches; we use the same batches in experiments to prevent additional gradient evaluations.

### 3.1 CONVERGENCE ANALYSIS

**Technical challenges.** The main difference of analyzing Algorithm 1 compared to other decentralized optimization algorithms is that the step size $\eta_{t,k}^i$ not only depends on the round $t$ and local steps $k$, *but also on $i$, the client*. To deal with the client-dependent step size, we require a slightly non-standard assumption on the dissimilarity between $f$ and $f_i$, as detailed below.

**Assumption 1.** *There exist nonnegative constants $\sigma, \rho$, and $G$ such that for all $i \in [M]$ and $x \in \mathbb{R}^d$,*

$$\mathbb{E}\|\nabla F_i(x, z) - \nabla f_i(x)\|^2 \leqslant \sigma^2, \qquad \text{(bounded variance)} \tag{1a}$$

$$\|\nabla f_i(x)\| \leqslant G, \qquad \text{(bounded gradient)} \tag{1b}$$

$$\|\nabla f_i(x) - \nabla f(x)\|^2 \leqslant \rho \|\nabla f(x)\|^2. \qquad \text{(strong growth of dissimilarity)} \tag{1c}$$

Assumption 1a has been standard in stochastic optimization literature (Ghadimi & Lan, 2013; Stich, 2018; Khaled et al., 2020); recently, this assumption has been relaxed to weaker ones such as expected smoothness (Gower et al., 2021), but this is out of scope of this work. Assumption 1b is fairly standard in nonconvex optimization (Zaheer et al., 2018; Ward et al., 2020), and often used in FL setting (Xie et al., 2019a;b; Reddi et al., 2021). Assumption 1c is reminiscent of the strong growth assumption in stochastic optimization (Schmidt & Roux, 2013), which is still used in recent works (Cevher & Vũ, 2019; Vaswani et al., 2019a;b). To the best of our knowledge, this is the first theoretical analysis of the FL setting where clients can use their own step size. We now present the main theorem.

**Theorem 1.** *Let Assumption 1 hold, with $\rho = \mathcal{O}(1)$. Further, suppose that $\gamma = \mathcal{O}(\frac{1}{K\sqrt{T}})$, and $\eta_0 = \mathcal{O}(\gamma)$. Then, the following property holds for Algorithm 1, for $T$ sufficiently large:*

$$\frac{1}{T} \sum_{t=0}^{T-1} \mathbb{E} \|\nabla f(x_t)\|^2 \leqslant \mathcal{O}\left(\frac{\Psi_1}{\sqrt{T}}\right) + \mathcal{O}\left(\frac{\tilde{L}^2 \Psi_2}{T}\right) + \mathcal{O}\left(\frac{\tilde{L}^3 \Psi_2}{\sqrt{T^3}}\right),$$

---

[2] For all our experiments, we use the default value $\gamma = 2$ from the original implementation in `https://github.com/ymalitsky/adaptive_GD/blob/master/pytorch/optimizer.py`.

where $\Psi_1 = \max\left\{\frac{\sigma^2}{b}, f(x_0) - f(x^\star)\right\}$ and $\Psi_2 = \left(\frac{\sigma^2}{b} + G^2\right)$ are global constants, with $b = |\mathcal{B}|$ being the batch size; $\tilde{L}$ is a constant at most the maximum of local smoothness, i.e., $\max_{i,t} \tilde{L}_{i,t}$, where $\tilde{L}_{i,t}$ the local smoothness of $f_i$ at round $t$.

The convergence result in Theorem 1 implies a sublinear convergence to an $\varepsilon$-first order stationary point with at least $T = \mathcal{O}(\varepsilon^{-2})$ communication rounds. We remind again that the conditions on $\gamma$ and $\eta_0$ are only required for our theory.[3] Importantly, we did not assume $f_i$ is globally $L$-smooth, a standard assumption in nonconvex optimization and FL literature (Ward et al., 2020; Koloskova et al., 2020; Li et al., 2020; Reddi et al., 2021); instead, we can obtain a smaller quantity, $\tilde{L}$, through our analysis; for space limitation, we defer the details and the proof to Appendix A.

## 4 EXPERIMENTAL SETUP AND RESULTS

We now introduce the experimental setup and discuss the results. Our implementation can be found in `https://github.com/jlylekim/auto-tuned-FL`.

**Datasets and models.** We consider image classification and text classification tasks. We use four datasets for image classification: MNIST, FMNIST, CIFAR-10, and CIFAR-100 (Krizhevsky et al., 2009). For MNIST and FMNIST, we train a shallow CNN with two convolutional and two fully-connected layers, followed by dropout and ReLU activations. For CIFAR-10, we train a ResNet-18 (He et al., 2016). For CIFAR-100, we train both ResNet-18 and ResNet-50 to study the effect of changing the model architecture. For text classification, we use two datasets: DBpedia and AGnews datasets (Zhang et al., 2015), and train a DistillBERT (Sanh et al., 2019) for classification.

For each dataset, we create a federated version by randomly partitioning the training data among 100 clients (50 for text classification), with each client getting 500 examples. To control the level of non-iidness, we apply latent Dirichlet allocation (LDA) over the labels following Hsu et al. (2019), where the degree class heterogeneity can be parameterized by the Dirichlet concentration parameter $\alpha$. The class distribution of different $\alpha$'s is visualized in Figure 8 in Appendix B.9.

**FL setup and optimizers.** We fix the number of clients to be 100 for the image classification, and 50 for the text classification; we randomly sample 10% as participating clients. We perform $E$ local epochs of training over each client's dataset. We utilize mini-batch gradients of size $b = 64$ for the image classification, and $b = 16$ for the text classification, leading to $K \approx \lfloor \frac{E \cdot 500}{b} \rfloor$ local gradient steps; we use $E = 1$ for all settings. For client optimizers, we compare stochastic gradient descent (SGD), SGD with momentum (SGDM), adaptive methods including Adam (Kingma & Ba, 2014), Adagrad (Duchi et al., 2011), SGD with stochastic Polyak step size (SPS) (Loizou et al., 2021), and our proposed method: $\Delta$-SGD in Algorithm 1. As our focus is on client adaptivity, we mostly present the results using FedAvg (McMahan et al., 2017) as the server optimizer; additional results using FedAdam can be found in Appendix B.4.

**Hyperparameters.** For each optimizer, we perform a grid search of learning rates on a *single task*: CIFAR-10 classification trained with a ResNet-18, with Dirichlet concentration parameter $\alpha = 0.1$; for the rest of the settings, we use the same learning rates. For SGD, we perform a grid search with $\eta \in \{0.01, 0.05, 0.1, 0.5\}$. For SGDM, we use the same grid for $\eta$ and use momentum parameter $\beta = 0.9$. To properly account for the SGD(M) fine-tuning typically done in practice, we also test dividing the step size by 10 after 50%, and again by 10 after 75% of the total training rounds (LR decay). For Adam and Adagrad, we grid search with $\eta \in \{0.001, 0.01, 0.1\}$. For SPS, we use the default setting of the official implementation.[4] For $\Delta$-SGD, we append $\delta$ in front of the second condition: $\sqrt{1 + \delta\theta^i_{t,k-1}\eta^i_{t,k-1}}$ following Malitsky & Mishchenko (2020), and use $\delta = 0.1$ for all experiments.[5] Finally, for the number of rounds $T$, we use 500 for MNIST, 1000 for FMNIST, 2000 for CIFAR-10 and CIFAR-100, and 100 for the text classification tasks.

---

[3]In all experiments, we use the default settings $\gamma = 2$, $\eta_0 = 0.2$, and $\theta_0 = 1$ without additional tuning.

[4]I.e., we use $f_i^\star = 0$, and $c = 0.5$, for the SPS step size: $\frac{f_i(x) - f_i^\star}{c\|\nabla f_i(x)\|^2}$. The official implementation can be found in `https://github.com/IssamLaradji/sps`.

[5]We also demonstrate in Appendix B.1 that $\delta$ has very little impact on the final accuracy of $\Delta$-SGD.

### 4.1 RESULTS

We clarify again that the best step size for each client optimizer was tuned via grid-search for a single task: CIFAR-10 classification trained with a ResNet-18, with Dirichlet concentration parameter $\alpha = 0.1$. We then intentionally use the same step size in all other tasks to highlight two points: $i$) $\Delta$-SGD works well without any tuning across different datasets, model architectures, and degrees of heterogeneity; $ii$) other optimizers perform suboptimally without additional tuning.

| Non-iidness | Optimizer | Dataset / Model | | | | |
|---|---|---|---|---|---|---|
| $\mathrm{Dir}(\alpha \cdot \mathbf{p})$ | | MNIST CNN | FMNIST CNN | CIFAR-10 ResNet-18 | CIFAR-100 ResNet-18 | CIFAR-100 ResNet-50 |
| $\alpha = 1$ | SGD | **98.3**$_{\downarrow(0.2)}$ | 86.5$_{\downarrow(0.8)}$ | 87.7$_{\downarrow(2.1)}$ | 57.7$_{\downarrow(4.2)}$ | 53.0$_{\downarrow(12.8)}$ |
| | SGD ($\downarrow$) | 97.8$_{\downarrow(0.7)}$ | 86.3$_{\downarrow(1.0)}$ | 87.8$_{\downarrow(2.0)}$ | **61.9**$_{\downarrow(0.0)}$ | 60.9$_{\downarrow(4.9)}$ |
| | SGDM | **98.5**$_{\downarrow(0.0)}$ | 85.2$_{\downarrow(2.1)}$ | 88.7$_{\downarrow(1.1)}$ | 58.8$_{\downarrow(3.1)}$ | 60.5$_{\downarrow(5.3)}$ |
| | SGDM ($\downarrow$) | **98.4**$_{\downarrow(0.1)}$ | **87.2**$_{\downarrow(0.1)}$ | 89.3$_{\downarrow(0.5)}$ | **61.4**$_{\downarrow(0.5)}$ | 63.3$_{\downarrow(2.5)}$ |
| | Adam | 94.7$_{\downarrow(3.8)}$ | 71.8$_{\downarrow(15.5)}$ | **89.4**$_{\downarrow(0.4)}$ | 55.6$_{\downarrow(6.3)}$ | 61.4$_{\downarrow(4.4)}$ |
| | Adagrad | 64.3$_{\downarrow(34.2)}$ | 45.5$_{\downarrow(41.8)}$ | 86.6$_{\downarrow(3.2)}$ | 53.5$_{\downarrow(8.4)}$ | 51.9$_{\downarrow(13.9)}$ |
| | SPS | 10.1$_{\downarrow(88.4)}$ | 85.9$_{\downarrow(1.4)}$ | 82.7$_{\downarrow(7.1)}$ | 1.0$_{\downarrow(60.9)}$ | 50.0$_{\downarrow(15.8)}$ |
| | $\Delta$-SGD | **98.4**$_{\downarrow(0.1)}$ | **87.3**$_{\downarrow(0.0)}$ | **89.8**$_{\downarrow(0.0)}$ | **61.5**$_{\downarrow(0.4)}$ | **65.8**$_{\downarrow(0.0)}$ |
| $\alpha = 0.1$ | SGD | **98.1**$_{\downarrow(0.0)}$ | 83.6$_{\downarrow(2.8)}$ | 72.1$_{\downarrow(12.9)}$ | 54.4$_{\downarrow(6.7)}$ | 44.2$_{\downarrow(19.9)}$ |
| | SGD ($\downarrow$) | **98.0**$_{\downarrow(0.1)}$ | 84.7$_{\downarrow(1.7)}$ | 78.4$_{\downarrow(6.6)}$ | 59.3$_{\downarrow(1.8)}$ | 48.7$_{\downarrow(15.4)}$ |
| | SGDM | **97.6**$_{\downarrow(0.5)}$ | 83.6$_{\downarrow(2.8)}$ | 79.6$_{\downarrow(5.4)}$ | 58.8$_{\downarrow(2.3)}$ | 52.3$_{\downarrow(11.8)}$ |
| | SGDM ($\downarrow$) | **98.0**$_{\downarrow(0.1)}$ | **86.1**$_{\downarrow(0.3)}$ | 77.9$_{\downarrow(7.1)}$ | 60.4$_{\downarrow(0.7)}$ | 52.8$_{\downarrow(11.3)}$ |
| | Adam | 96.4$_{\downarrow(1.7)}$ | 80.4$_{\downarrow(6.0)}$ | **85.0**$_{\downarrow(0.0)}$ | 55.4$_{\downarrow(5.7)}$ | 58.2$_{\downarrow(5.9)}$ |
| | Adagrad | 89.9$_{\downarrow(8.2)}$ | 46.3$_{\downarrow(40.1)}$ | 84.1$_{\downarrow(0.9)}$ | 49.6$_{\downarrow(11.5)}$ | 48.0$_{\downarrow(16.1)}$ |
| | SPS | 96.0$_{\downarrow(2.1)}$ | 85.0$_{\downarrow(1.4)}$ | 70.3$_{\downarrow(14.7)}$ | 42.2$_{\downarrow(18.9)}$ | 42.2$_{\downarrow(21.9)}$ |
| | $\Delta$-SGD | **98.1**$_{\downarrow(0.0)}$ | **86.4**$_{\downarrow(0.0)}$ | **84.5**$_{\downarrow(0.5)}$ | **61.1**$_{\downarrow(0.0)}$ | **64.1**$_{\downarrow(0.0)}$ |
| $\alpha = 0.01$ | SGD | 96.8$_{\downarrow(0.7)}$ | 79.0$_{\downarrow(1.2)}$ | 22.6$_{\downarrow(11.3)}$ | 30.5$_{\downarrow(1.3)}$ | 24.3$_{\downarrow(7.1)}$ |
| | SGD ($\downarrow$) | **97.2**$_{\downarrow(0.3)}$ | 79.3$_{\downarrow(0.9)}$ | **33.9**$_{\downarrow(0.0)}$ | 30.3$_{\downarrow(1.5)}$ | 24.6$_{\downarrow(6.8)}$ |
| | SGDM | 77.9$_{\downarrow(19.6)}$ | 75.7$_{\downarrow(4.5)}$ | 28.4$_{\downarrow(5.5)}$ | 24.8$_{\downarrow(7.0)}$ | 22.0$_{\downarrow(9.4)}$ |
| | SGDM ($\downarrow$) | 94.0$_{\downarrow(3.5)}$ | 79.5$_{\downarrow(0.7)}$ | 29.0$_{\downarrow(4.9)}$ | 20.9$_{\downarrow(10.9)}$ | 14.7$_{\downarrow(16.7)}$ |
| | Adam | 80.8$_{\downarrow(16.7)}$ | 60.6$_{\downarrow(19.6)}$ | 22.1$_{\downarrow(11.8)}$ | 18.2$_{\downarrow(13.6)}$ | 22.6$_{\downarrow(8.8)}$ |
| | Adagrad | 72.4$_{\downarrow(25.1)}$ | 45.9$_{\downarrow(34.3)}$ | 12.5$_{\downarrow(21.4)}$ | 25.8$_{\downarrow(6.0)}$ | 22.2$_{\downarrow(9.2)}$ |
| | SPS | 69.7$_{\downarrow(27.8)}$ | 44.0$_{\downarrow(36.2)}$ | 21.5$_{\downarrow(12.4)}$ | 22.0$_{\downarrow(9.8)}$ | 17.4$_{\downarrow(14.0)}$ |
| | $\Delta$-SGD | **97.5**$_{\downarrow(0.0)}$ | **80.2**$_{\downarrow(0.0)}$ | 31.6$_{\downarrow(2.3)}$ | **31.8**$_{\downarrow(0.0)}$ | **31.4**$_{\downarrow(0.0)}$ |

Table 1: *Experimental results based on the settings detailed in Section 4.* Best accuracy ($\pm$ 0.5%) for each task are shown in **bold**. Subscripts $_{\downarrow(x.x)}$ is the performance difference from the best result, and is highlighted in pink when it is bigger than 2%. The down-arrow symbol ($\downarrow$) indicates step-wise learning rate decay, where the step sizes are divided by 10 after 50%, and another by 10 after 75% of the total rounds.

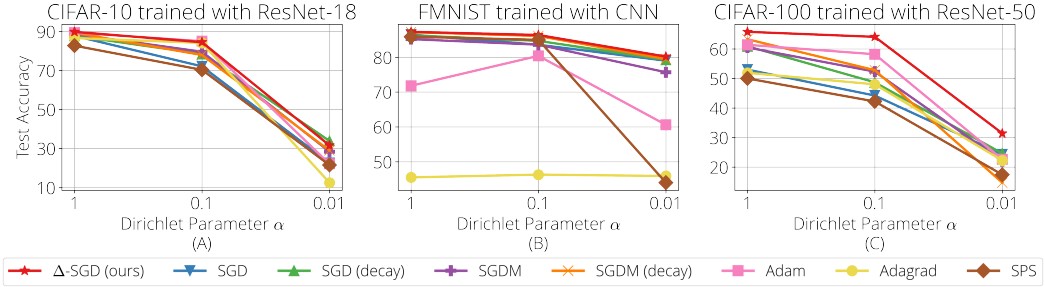

Figure 2: *The effect of stronger heterogeneity on different client optimizers, induced by the Dirichlet concentration parameter $\alpha \in \{0.01, 0.1, 1\}$.* $\Delta$-SGD remains robust performance in all cases, whereas other methods show significant performance degradation when changing the level of heterogeneity $\alpha$, or when changing the setting (model/architecture).

**Changing the level of non-iidness.** We first investigate how the performance of different client optimizers degrade in increasing degrees of heterogeneity by varying the concentration parameter $\alpha \in \{1, 0.1, 0.01\}$ multiplied to the prior of Dirichlet distribution, following Hsu et al. (2019).

Three illustrative cases are visualized in Figure 2. We remind that the step sizes are tuned for the task in Figure 2(A). For this task, with $\alpha = 1$ (i.e., closer to iid), all methods perform better than $\alpha = 0.1$, as expected. With $\alpha = 0.01$, which is highly non-iid (c.f., Figure 8 in Appendix B.9), we see a significant drop in performance for all methods. SGD with LR decay and $\Delta$-SGD perform the best, while adaptive methods like Adam ($85\% \rightarrow 22.1\%$) and Adagrad ($84.1\% \rightarrow 12.5\%$) degrade noticeably more than other methods.

Figure 2(B) shows the results of training FMNIST with a CNN. This problem is generally considered easier than CIFAR-10 classification. The experiments show that the performance degradation with varying $\alpha$'s is much milder in this case. Interestingly, adaptive methods such as Adam, Adagrad, and SPS perform much worse than other methods

Lastly, in Figure 2(C), results for CIFAR-100 classification trained with a ResNet-50 are illustrated. $\Delta$-SGD exhibits superior performance in all cases of $\alpha$. Unlike MNIST and FMNIST, Adam enjoys the second ($\alpha = 0.1$) or the third ($\alpha = 1$) best performance in this task, complicating how one should "tune" Adam for the task at hand. Other methods, including SGD with and without momentum/LR decay, Adagrad, and SPS, perform much worse than $\Delta$-SGD.

**Changing the model architecture/dataset.** For this remark, let us focus on CIFAR-100 trained on ResNet-18 versus on ResNet-50, with $\alpha = 0.1$, illustrated in Figure 3(A). SGD and SGDM (both with and without LR decay), Adagrad, and SPS perform worse using ResNet-50 than ResNet-18. This is a *counter-intuitive* behavior, as one would expect to get better accuracy by using a more powerful model, although the step sizes are tuned on ResNet-18. $\Delta$-SGD is an exception: without any additional tuning, $\Delta$-*SGD can improve its performance*. Adam also improves similarly, but the achieved accuracy is significantly worse than that of $\Delta$-SGD.

We now focus on cases where the dataset changes, but the model architecture remains the same. We mainly consider two cases, illustrated in Figure 3(B) and (C): When CNN is trained for classifying MNIST versus FMNIST, and when ResNet-18 is trained for CIFAR-10 versus CIFAR-100. From (B), one can infer the difficulty in tuning SGDM: without the LR decay, SGDM only achieves around 78% accuracy for MNIST classification, while SGD without LR decay still achieves over 96% accuracy. For FMNIST on the other hand, SGDM performs relatively well, but adaptive methods like Adam, Adagrad, and SPS degrade significantly.

Aggravating the complexity of fine-tuning SGD(M) for MNIST, one can observe from (C) a similar trend in CIFAR-10 trained with ResNet-18. In that case, $\Delta$-SGD does not achieve the best test accuracy (although it is the second best with a pretty big margin with the rest), while SGD with LR decay does. However, without LR decay, the accuracy achieved by SGD drops *more than* 11%. Again, adaptive methods like Adam, Adagrad, and SPS perform suboptimally in this case.

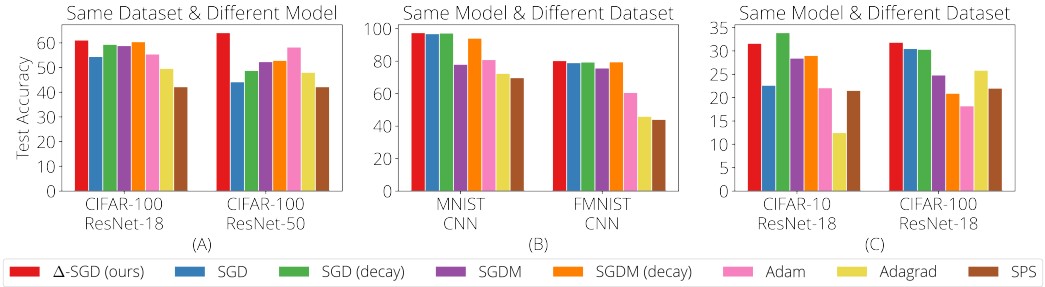

Figure 3: *The effect of changing the dataset and the model architecture on different client optimizers.* $\Delta$-SGD remains superior performance without additional tuning when model or dataset changes, whereas other methods often degrades in performance. (A): CIFAR-100 trained with ResNet-18 versus Resnet-50 ($\alpha = 0.1$), (B): MNIST versus FMNIST trained with CNN ($\alpha = 0.01$), (C): CIFAR-10 versus CIFAR-100 trained with ResNet-18 ($\alpha = 0.01$).

**Changing the domain: text classification.** In Table 2a, we make a bigger leap, and test the performance of different client optimizers for text classification tasks. We use the DBpedia and AGnews datasets (Zhang et al., 2015), and train a DistillBERT (Sanh et al., 2019) for classification. Again, the best step size for each client optimizer was tuned for the CIFAR-10 image classification trained with a ResNet-18, for $\alpha = 0.1$. Thus, the type of the dataset changed completely from image to text, as well as the model architecture: from vision to language.

Not surprisingly, four methods (SGDM, SGDM($\downarrow$), Adam, and Adagrad) ended up learning nothing (i.e., the achieved accuracy is 1/(# of labels)), indicating the fine-tuned step sizes for these methods using CIFAR-10 classification task was too large for the considered text classification tasks. $\Delta$-SGD, on the other hand, still achieves competitive accuracies without additional tuning, thanks to the locality adaptive step size. Interestingly, SGD, SGD($\downarrow$), and SPS worked well for the text classification tasks, which is contrary to their suboptimal performances for image classification tasks in Table 1.

**Changing the loss function.** In Table 2b, we illustrate how $\Delta$-SGD performs when combined with a different loss function, such as FedProx (Li et al., 2020). FedProx aims to address heterogeneity; we thus used the lowest concentration parameter, $\alpha = 0.01$. We only present a subset of results: CIFAR-10(100) classification with ResNet-18(50) (additional results in Appendix B.6). The results show that $\Delta$-SGD again outperforms all other methods with a significant margin: $\Delta$-SGD is not only the best, but also the performance difference is at least 2.5%, and as large as 26.9%. Yet, compared to the original setting without the proximal term in Table 1, it is unclear whether or not the additional proximal term helps. For $\Delta$-SGD, the accuracy increased for CIFAR-10 with ResNet-18, but slightly got worse for CIFAR-100 with ResNet-50; a similar story applies to other methods.

| Text classification | *Dataset / Model* | |
| --- | --- | --- |
| $\alpha = 1$ | Agnews | Dbpedia |
| *Optimizer* | DistillBERT | |
| SGD | $91.1_{\downarrow(0.5)}$ | $96.0_{\downarrow(2.9)}$ |
| SGD ($\downarrow$) | $91.6_{\downarrow(0.0)}$ | $98.7_{\downarrow(0.2)}$ |
| SGDM | $25.0_{\downarrow(66.6)}$ | $7.1_{\downarrow(91.8)}$ |
| SGDM ($\downarrow$) | $25.0_{\downarrow(66.6)}$ | $7.1_{\downarrow(91.8)}$ |
| Adam | $25.0_{\downarrow(66.6)}$ | $7.1_{\downarrow(91.8)}$ |
| Adagrad | $25.0_{\downarrow(66.6)}$ | $7.1_{\downarrow(91.8)}$ |
| SPS | $91.5_{\downarrow(0.1)}$ | $98.9_{\downarrow(0.0)}$ |
| $\Delta$-SGD | $90.7_{\downarrow(0.9)}$ | $98.6_{\downarrow(0.3)}$ |

| FedProx | *Dataset / Model* | |
| --- | --- | --- |
| $\alpha = 0.01$ | CIFAR-10 | CIFAR-100 |
| *Optimizer* | ResNet-18 | ResNet-50 |
| SGD | $20.0_{\downarrow(13.8)}$ | $25.2_{\downarrow(5.9)}$ |
| SGD ($\downarrow$) | $31.3_{\downarrow(2.5)}$ | $20.2_{\downarrow(10.8)}$ |
| SGDM | $29.3_{\downarrow(4.4)}$ | $23.8_{\downarrow(7.2)}$ |
| SGDM ($\downarrow$) | $25.3_{\downarrow(8.5)}$ | $15.0_{\downarrow(16.0)}$ |
| Adam | $28.1_{\downarrow(5.7)}$ | $22.6_{\downarrow(8.4)}$ |
| Adagrad | $19.3_{\downarrow(14.5)}$ | $4.1_{\downarrow(26.9)}$ |
| SPS | $27.6_{\downarrow(6.2)}$ | $16.5_{\downarrow(14.5)}$ |
| $\Delta$-SGD | $33.8_{\downarrow(0.0)}$ | $31.0_{\downarrow(0.0)}$ |

(a) *Experimental results for text classification*    (b) *Experimental results using **FedProx** loss.*

Table 2: Additional experiments for a different domain (a), and a different loss function (b).

**Other findings.** We present all the omitted theorems and proofs in Appendix A, and additional empirical findings in Appendix B. Specifically, we demonstrate: the robustness of the performance of $\Delta$-SGD (Appendix B.1), the effect of the different numbers of local iterations (Appendix B.2), additional experiments when the size of local dataset is different per client (Appendix B.3), additional experimental results using FedAdam (Reddi et al., 2021) (Appendix B.4), additional experiential results using MOON (Appendix B.5) and FedProx (Appendix B.6) loss functions, and three independent trials for a subset of tasks (Appendix B.7). We also visualize the step size conditions for $\Delta$-SGD (Appendix B.8) as well as the degree of heterogeneity (Appendix B.9).

## 5 CONCLUSION

In this work, we proposed $\Delta$-SGD, a distributed SGD scheme equipped with an adaptive step size that enables each client to use its own step size and adapts to the local smoothness of the function each client is optimizing. We proved the convergence of $\Delta$-SGD in the general nonconvex setting and presented extensive empirical results, where the superiority of $\Delta$-SGD is shown in various scenarios without any tuning. For future works, extending $\Delta$-SGD to a coordinate-wise step size in the spirit of Duchi et al. (2011); Kingma & Ba (2014), applying the step size to more complex algorithms like (local) factored gradient descent (Kim et al., 2022a), and enabling asynchronous updates (Assran et al., 2020; Toghani et al., 2022; Nguyen et al., 2022) could be interesting directions.

ACKNOWLEDGMENTS

This work is supported by NSF FET: Small No. 1907936, NSF MLWiNS CNS No. 2003137 (in collaboration with Intel), NSF CMMI No. 2037545, NSF CAREER award No. 2145629, NSF CIF No. 2008555, Rice InterDisciplinary Excellence Award (IDEA), NSF CCF No. 2211815, NSF DEB No. 2213568, and Lodieska Stockbridge Vaughn Fellowship.

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

SUPPLEMENTARY MATERIALS FOR
"ADAPTIVE FEDERATED LEARNING WITH AUTO-TUNED CLIENTS"

In this appendix, we provide all missing proofs that were not present in the main text, as well as additional plots and experiments. The appendix is organized as follows.

- In Appendix A, we provide all the missing proofs. In particular:
  - In Section A.1, we provide the proof of Algorithm 1, under Assumption 1.
  - In Section A.2, we provide the proof of Algorithm, 1, with some modified assumptions, including that $f_i$ is convex for all $i \in [m]$.
- In Appendix B, we provide additional experiments, as well as miscellaneous plots that were missing in the main text due to the space constraints. Specifically:
  - In Section B.1, we provide some perspectives on the ease of tuning of $\Delta$-SGD in Algorithm 1.
  - In Section B.2, we provide some perspectives on how $\Delta$-SGD performs using different number of local iterations.
  - In Section B.3, we provide additional experimental results where each client has different number of samples as local dataset.
  - In Section B.4, we provide additional experimental results using FedAdam (Reddi et al., 2021) server-side adaptive method.
  - In Section B.5, we provide additional experimental results using MOON (Li et al., 2021) loss function.
  - In Section B.6, we provide additional experiemtns using FedProx (Li et al., 2020) loss function.
  - In Section B.7, we perform three independent trials for a subset of tasks for all client-optimizers. We plot the average and the standard deviations in Figure 6, and report those values in Table 7.
  - In Section B.8, we visualize the step size conditions for $\Delta$-SGD.
  - In Section B.9, we provide illustration of the different level of heterogeneity induced by the Dirichlet concentration parameter $\alpha$.

## A  MISSING PROOFS

We first introduce the following two inequalities, which will be used in the subsequent proofs:

$$\langle x_i, x_j \rangle \leqslant \frac{1}{2}\|x_i\|^2 + \frac{1}{2}\|x_j\|^2, \quad \text{and} \tag{7}$$

$$\Big\| \sum_{i=1}^{m} x_i \Big\|^2 \leqslant m \sum_{i=1}^{m} \Big\| x_i \Big\|^2, \tag{8}$$

for any set of $m$ vectors $\{x_i\}_{i=1}^m$ with $x_i \in \mathbb{R}^d$.

We also provide the following lemma, which establishes $\tilde{L}$-smoothness based on the local smoothness.

**Lemma 2.** *For locally smooth function $f_i$, the following inequality holds:*

$$f_i(y) \leqslant f_i(x) + \langle \nabla f_i(x), y - x \rangle + \frac{\tilde{L}_{i,t}}{2}\|y - x\|^2 \quad \forall x, y \in \mathcal{S}_{i,t}, \tag{9}$$

*where $\mathcal{S}_{i,t} := \text{conv}(\{x_{t,k}^i\}_{k=1}^K \cup \{x_t, x_{t+1}\})$.*

*Proof.* First, notice that by Assumptions (1a) and (1b), stochastic gradients are also bounded for all $x \in \mathbb{R}^d$. Therefore, the local iterates, i.e., $\{x_{t,k}^i\}_{k=0}^K$ remain bounded. Hence, the set $\mathcal{S}_{i,t} := \text{conv}(\{x_{t,k}^i\}_{k=1}^K \cup \{x_t, x_{t+1}\})$, where $\text{conv}(\cdot)$ denotes the convex hull, is bounded. Due to the fact

that $f_i$ is locally smooth, $\nabla f$ is Lipschitz continuous on bounded sets. This implies there exists a positive constant $\tilde{L}_{i,t}$ such that,

$$\|\nabla f_i(x) - \nabla f_i(y)\| \leqslant \tilde{L}_{i,t}\|y - x\| \quad \forall x, y \in \mathcal{S}_{i,t}. \tag{10}$$

Thus, we have

$$\langle \nabla f_i(y) - \nabla f_i(x), y - x \rangle \leqslant \|\nabla f_i(y) - \nabla f_i(x)\| \cdot \|y - x\|$$
$$\leqslant \tilde{L}_{i,t}\|y - x\|^2, \quad \forall x, y \in \mathcal{S}_{i,t}.$$

Let $g(\tau) = f_i(x + \tau(y - x))$. Then, we have

$$g'(\tau) = \langle \nabla f_i(x + \tau(y - x)), y - x \rangle$$
$$g'(\tau) - g'(0) = \langle \nabla f_i(x + \tau(y - x)) - \nabla f_i(x), y - x \rangle$$
$$\leqslant \|\nabla f_i(x + \tau(y - x)) - \nabla f_i(x)\| \cdot \|y - x\|$$
$$\leqslant \tau \tilde{L}_{i,t}\|y - x\|^2.$$

Using the mean value theorem, we have

$$f_i(y) = g(1) = g(0) + \int_0^1 g'(\tau)d\tau$$
$$\leqslant g(0) + \int_0^1 [g'(0) + \tau \tilde{L}_{i,t}\|y - x\|^2]d\tau$$
$$\leqslant g(0) + g'(0) + \frac{\tilde{L}_{i,t}}{2}\|y - x\|^2$$
$$\leqslant f_i(x) + \langle \nabla f_i(x), y - x \rangle + \frac{\tilde{L}_{i,t}}{2}\|y - x\|^2.$$

$\square$

Now, we define $\tilde{L}_i := \max_t \tilde{L}_{i,t}$. Given this definition, we relate the local-smoothness constant $\tilde{L}$ of the average function $f$ with (individual) local-smoothness constants $\tilde{L}_i$ in the below lemma.

**Lemma 3.** *The average of $\tilde{L}_i$-local smooth functions $f_i$ are at least $(\frac{1}{m}\sum_{i=1}^m \tilde{L}_i)$-local smooth.*

*Proof.* Starting from (10), we have:

$$\left\|\frac{1}{m}\sum_{i=1}^m \nabla f_i(x) - \frac{1}{m}\sum_{i=1}^m \nabla f_i(y)\right\| = \left\|\frac{1}{m}\sum_{i=1}^m (\nabla f_i(x) - \nabla f_i(y))\right\|$$
$$\leqslant \frac{1}{m}\sum_{i=1}^m \|\nabla f_i(x) - \nabla f_i(y)\|$$
$$\leqslant \frac{1}{m}\sum_{i=1}^m \tilde{L}_i\|x - y\| \tag{11}$$
$$\leqslant \tilde{L}\|x - y\|, \quad \text{where} \quad \tilde{L} = \max_{i,t} \tilde{L}_{i,t}. \tag{12}$$

Technically, (11) can provide a slightly tighter bound for Theorem 1, but (12) simplifies the final expression. We also have the following form:

$$f(x_{t+1}) \leqslant f(x_t) + \langle \nabla f(x_t), x_{t+1} - x_t \rangle + \frac{\tilde{L}}{2}\|x_{t+1} - x_t\|^2,$$

which can be obtained from Lemma 2. $\square$

A.1   PROOF OF ALGORITHM 1 UNDER ASSUMPTION 1

*Proof.* Based on Algorithm 1, we use the following notations throughout the proof:

$$x_{t,0}^i = x_t,$$
$$x_{t,k}^i = x_{t,k-1}^i - \eta_{t,k}^i \tilde{\nabla} f_i(x_{t,k-1}^i), \quad \forall k \in [K],$$
$$x_{t+1} = \frac{1}{|\mathcal{S}_t|} \sum_{i \in \mathcal{S}_t} x_{t,k}^i,$$

where we denoted with $x_{t+1}$ as the server parameter at round $t+1$, which is the average of the local parameters $x_{t,k}^i$ for all client $\forall\, i \in [m]$ after running $k$ local iterations for $\forall\, k \in [K]$.

We also recall that we use $\tilde{\nabla} f_i(x) = \frac{1}{|\mathcal{B}|} \sum_{z \in \mathcal{B}} \nabla F_i(x, z)$ to denote the stochastic gradients with batch size $|\mathcal{B}| = b$. Note that whenever we use this shorthand notation in the theory, we imply that we draw a new independent batch of samples to compute the stochastic gradients.

We start from the smoothness of $f(\cdot)$, which we note again is obtained via Lemma 3, i.e., as a byproduct of our analysis, and we do not assume that $f_i$'s are *globally* $L$-smooth. Also note that Lemma 3 holds for $\{x_0, x_1, \dots\}$, i.e., the average of the local parameters visited by Algorithm 1.

Proceeding, we have

$$f(x_{t+1}) \leqslant f(x_t) - \Big\langle \nabla f(x_t), \frac{1}{|\mathcal{S}_t|} \sum_{i \in \mathcal{S}_t} \sum_{k=0}^{K-1} \eta_{t,k}^i \tilde{\nabla} f_i(x_{t,k}^i) \Big\rangle + \frac{\tilde{L}}{2} \Big\| \frac{1}{|\mathcal{S}_t|} \sum_{i \in \mathcal{S}_t} \sum_{k=0}^{K-1} \eta_{t,k}^i \tilde{\nabla} f_i(x_{t,k}^i) \Big\|^2.$$

Taking expectation from the expression above, where we overload the notation such that the expectation is both with respect to the client sampling randomness (indexed with $i$) as well as the data sampling randomness (indexed with $z$), that is $\mathbb{E}(\cdot) := \mathbb{E}_{\mathcal{F}_t}\left[\mathbb{E}_i\left[\mathbb{E}_z[\cdot \mid \mathcal{F}_t, i]|\mathcal{F}_t]\right]\right]$, where $\mathcal{F}_t$ is the natural filtration, we have:

$$\mathbb{E} f(x_{t+1})$$

$$\leqslant f(x_t) - \mathbb{E}\Big\langle \nabla f(x_t), \frac{1}{|\mathcal{S}_t|} \sum_{i \in \mathcal{S}_t} \sum_{k=0}^{K-1} \eta_{t,k}^i \tilde{\nabla} f_i(x_{t,k}^i) \Big\rangle + \frac{\tilde{L}}{2} \mathbb{E}\Big\| \frac{1}{|\mathcal{S}_t|} \sum_{i \in \mathcal{S}_t} \sum_{k=0}^{K-1} \eta_{t,k}^i \tilde{\nabla} f_i(x_{t,k}^i) \Big\|^2 \quad (13)$$

$$\leqslant f(x_t) - \mathbb{E}\Big\langle \nabla f(x_t), \frac{1}{m} \sum_{i=1}^{m} \sum_{k=0}^{K-1} \eta_{t,k}^i \nabla f_i(x_{t,k}^i) \Big\rangle + \frac{\tilde{L}}{2m} \sum_{i=1}^{m} \mathbb{E}\Big\| \sum_{k=0}^{K-1} \eta_{t,k}^i \tilde{\nabla} f_i(x_{t,k}^i) \Big\|^2 \quad (14)$$

$$= f(x_t) - D_t \|\nabla f(x_t)\|^2 - D_t \mathbb{E}\Big\langle \nabla f(x_t), \frac{1}{m} \sum_{i=1}^{m} \sum_{k=0}^{K-1} \frac{\eta_{t,k}^i}{D_t} \big(\nabla f_i(x_{t,k}^i) - \nabla f(x_t)\big) \Big\rangle$$

$$+ \frac{\tilde{L}}{2m} \sum_{i=1}^{m} \mathbb{E}\Big\| \sum_{k=0}^{K-1} \eta_{t,k}^i \tilde{\nabla} f_i(x_{t,k}^i) \Big\|^2, \quad (15)$$

where in the equality, we added and subtracted $\frac{1}{m} \sum_{i=1}^{m} \sum_{k=0}^{K-1} \eta_{t,k}^i \nabla f(x_t)$ in the inner product term, and also replaced $D_t := \frac{1}{m} \sum_{i=1}^{m} \sum_{k=0}^{K-1} \eta_{t,k}^i$. Then, using (7) and (13)-(15), we have

$$\mathbb{E} f(x_{t+1}) \leqslant f(x_t) - D_t \|\nabla f(x_t)\|^2 + \frac{D_t}{2} \|\nabla f(x_t)\|^2 \quad (16)$$

$$+ \frac{D_t}{2} \mathbb{E}\Big\| \frac{1}{m} \sum_{i=1}^{m} \sum_{k=0}^{K-1} \frac{\eta_{t,k}^i}{D_t} \big(\nabla f_i(x_{t,k}^i) - \nabla f(x_t)\big) \Big\|^2 + \frac{\tilde{L}}{2m} \sum_{i=1}^{m} \mathbb{E}\Big\| \sum_{k=0}^{K-1} \eta_{t,k}^i \tilde{\nabla} f_i(x_{t,k}^i) \Big\|^2$$

$$= f(x_t) - \frac{D_t}{2} \|\nabla f(x_t)\|^2 + \frac{1}{2D_t} \mathbb{E}\Big\| \frac{1}{m} \sum_{i=1}^{m} \sum_{k=0}^{K-1} \eta_{t,k}^i \big(\nabla f_i(x_{t,k}^i) - \nabla f(x_t)\big) \Big\|^2$$

$$+ \frac{\tilde{L}}{2m} \sum_{i=1}^{m} \mathbb{E}\Big\| \sum_{k=0}^{K-1} \eta_{t,k}^i \tilde{\nabla} f_i(x_{t,k}^i) \Big\|^2 \quad (17)$$

$$= f(x_t) - \frac{D_t}{2} \left\| \nabla f(x_t) \right\|^2$$

$$+ \frac{1}{2D_t} \mathbb{E} \Big\| \frac{1}{m} \sum_{i=1}^{m} \sum_{k=0}^{K-1} \eta_{t,k}^i \Big( \nabla f_i(x_t) - \nabla f_i(x_{t,k}^i) + \nabla f_i(x_t) - \nabla f(x_t) \Big) \Big\|^2$$

$$+ \frac{\tilde{L}}{2m} \sum_{i=1}^{m} \mathbb{E} \Big\| \sum_{k=0}^{K-1} \eta_{t,k}^i \Big( \tilde{\nabla} f_i(x_{t,k}^i) - \nabla f_i(x_{t,k}^i) + \nabla f_i(x_{t,k}^i) - \nabla f_i(x_t)$$

$$+ \nabla f_i(x_t) - \nabla f(x_t) + \nabla f(x_t) \Big) \Big\|^2 \tag{18}$$

We now expand the terms in (18) and bound each term separately, as follows.

$$\mathbb{E} f(x_{t+1}) \leqslant f(x_t) - \frac{D_t}{2} \left\| \nabla f(x_t) \right\|^2$$

$$+ \underbrace{\frac{1}{D_t} \mathbb{E} \Big\| \frac{1}{m} \sum_{i=1}^{m} \sum_{k=0}^{K-1} \eta_{t,k}^i \Big( \nabla f_i(x_t) - \nabla f_i(x_{t,k}^i) \Big) \Big\|^2}_{A_1}$$

$$+ \underbrace{\frac{1}{D_t} \mathbb{E} \Big\| \frac{1}{m} \sum_{i=1}^{m} \sum_{k=0}^{K-1} \eta_{t,k}^i \Big( \nabla f_i(x_t) - \nabla f(x_t) \Big) \Big\|^2}_{A_2}$$

$$+ \underbrace{\frac{2\tilde{L}}{m} \sum_{i=1}^{m} \mathbb{E} \Big\| \sum_{k=0}^{K-1} \eta_{t,k}^i \Big( \tilde{\nabla} f_i(x_{t,k}^i) - \nabla f_i(x_{t,k}^i) \Big) \Big\|^2}_{A_3}$$

$$+ \underbrace{\frac{2\tilde{L}}{m} \sum_{i=1}^{m} \mathbb{E} \Big\| \sum_{k=0}^{K-1} \eta_{t,k}^i \Big( \nabla f_i(x_{t,k}^i) - \nabla f_i(x_t) \Big) \Big\|^2}_{A_4}$$

$$+ \underbrace{\frac{2\tilde{L}}{m} \sum_{i=1}^{m} \mathbb{E} \Big\| \sum_{k=0}^{K-1} \eta_{t,k}^i \Big( \nabla f_i(x_t) - \nabla f(x_t) \Big) \Big\|^2}_{A_5}$$

$$+ \underbrace{\frac{2\tilde{L}}{m} \sum_{i=1}^{m} \mathbb{E} \Big\| \sum_{k=0}^{K-1} \eta_{t,k}^i \nabla f(x_t) \Big\|^2}_{A_6}. \tag{19}$$

Now, we provide an upper bound for each term in (19). Starting with $A_1$, using (7), (8), and (12), we have

$$A_1 \leqslant \frac{K}{D_t m} \sum_{i=1}^{m} \sum_{k=0}^{K-1} (\eta_{t,k}^i)^2 \, \mathbb{E} \Big\| \nabla f_i(x_t) - \nabla f_i(x_{t,k}^i) \Big\|^2 \tag{20}$$

$$\leqslant \frac{K \tilde{L}^2}{D_t m} \sum_{i=1}^{m} \sum_{k=0}^{K-1} (\eta_{t,k}^i)^2 \, \mathbb{E} \Big\| x_t - x_{t,k}^i \Big\|^2$$

$$= \frac{K \tilde{L}^2}{D_t m} \sum_{i=1}^{m} \sum_{k=0}^{K-1} (\eta_{t,k}^i)^2 \, \mathbb{E} \Big\| \sum_{\ell=0}^{k-1} \eta_{t,l}^i \tilde{\nabla} f_i(x_{t,\ell}^i) \Big\|^2$$

$$= \frac{K \tilde{L}^2}{D_t m} \sum_{i=1}^{m} \sum_{k=0}^{K-1} (\eta_{t,k}^i)^2 \, \mathbb{E} \Big\| \sum_{\ell=0}^{k-1} \eta_{t,l}^i \Big( \tilde{\nabla} f_i(x_{t,\ell}^i) - \nabla f_i(x_{t,\ell}^i) + \nabla f_i(x_{t,\ell}^i) \Big) \Big\|^2$$

$$\leqslant \frac{2K\tilde{L}^2}{D_t m} \sum_{i=1}^{m} \sum_{k=0}^{K-1} (\eta_{t,k}^i)^2 \left[ \mathbb{E} \left\| \sum_{\ell=0}^{k-1} \eta_{t,l}^i \Big( \tilde{\nabla} f_i(x_{t,\ell}^i) - \nabla f_i(x_{t,\ell}^i) \Big) \right\|^2 + \mathbb{E} \left\| \sum_{\ell=0}^{k-1} \eta_{t,l}^i \nabla f_i(x_{t,\ell}^i) \right\|^2 \right]$$

$$\leqslant \frac{2K\tilde{L}^2}{D_t m} \sum_{i=1}^{m} \sum_{k=0}^{K-1} k(\eta_{t,k}^i)^2 \sum_{\ell=0}^{k-1} (\eta_{t,l}^i)^2 \left[ \mathbb{E} \left\| \tilde{\nabla} f_i(x_{t,\ell}^i) - \nabla f_i(x_{t,\ell}^i) \right\|^2 + \mathbb{E} \left\| \nabla f_i(x_{t,\ell}^i) \right\|^2 \right]$$

$$\leqslant \frac{2K\tilde{L}^2}{D_t m} \sum_{i=1}^{m} \sum_{k=0}^{K-1} k(\eta_{t,k}^i)^2 \sum_{\ell=0}^{k-1} (\eta_{t,l}^i)^2 \left( \frac{\sigma^2}{b} + G^2 \right)$$

$$\leqslant \frac{2K^2\tilde{L}^2 \left( \frac{\sigma^2}{b} + G^2 \right)}{D_t m} \sum_{i=1}^{m} \left( \sum_{k=0}^{K-1} (\eta_{t,k}^i)^2 \right)^2. \tag{21}$$

For $A_2$, using (7) and (8), we have:

$$A_2 \leqslant \frac{K}{D_t m} \sum_{i=1}^{m} \sum_{k=0}^{K-1} (\eta_{t,k}^i)^2 \, \mathbb{E} \left\| \nabla f_i(x_t) - \nabla f(x_t) \right\|^2$$

$$\leqslant \frac{K\rho}{D_t m} \left\| \nabla f(x_t) \right\|^2 \sum_{i=1}^{m} \sum_{k=0}^{K-1} (\eta_{t,k}^i)^2. \tag{22}$$

For $A_3$, using (7), (8), and (12), we have:

$$A_3 \leqslant \frac{2K\tilde{L}}{m} \sum_{i=1}^{m} \sum_{k=0}^{K-1} (\eta_{t,k}^i)^2 \, \mathbb{E} \left\| \tilde{\nabla} f_i(x_{t,k}^i) - \nabla f_i(x_{t,k}^i) \right\|^2$$

$$\leqslant \frac{2K\tilde{L}\sigma^2}{mb} \sum_{i=1}^{m} \sum_{k=0}^{K-1} (\eta_{t,k}^i)^2. \tag{23}$$

For $A_4$, using (7), (8), and (12), we have:

$$A_4 \leqslant \frac{2K\tilde{L}}{m} \sum_{i=1}^{m} \sum_{k=0}^{K-1} (\eta_{t,k}^i)^2 \, \mathbb{E} \left\| \nabla f_i(x_t) - \nabla f_i(x_{t,k}^i) \right\|^2 \tag{24}$$

$$\leqslant \frac{2K\tilde{L}^3}{m} \sum_{i=1}^{m} \sum_{k=0}^{K-1} (\eta_{t,k}^i)^2 \, \mathbb{E} \left\| x_t - x_{t,k}^i \right\|^2$$

$$= \frac{2K\tilde{L}^3}{m} \sum_{i=1}^{m} \sum_{k=0}^{K-1} (\eta_{t,k}^i)^2 \, \mathbb{E} \left\| \sum_{\ell=0}^{k-1} \eta_{t,l}^i \tilde{\nabla} f_i(x_{t,\ell}^i) \right\|^2$$

$$= \frac{2K\tilde{L}^3}{m} \sum_{i=1}^{m} \sum_{k=0}^{K-1} (\eta_{t,k}^i)^2 \, \mathbb{E} \left\| \sum_{\ell=0}^{k-1} \eta_{t,l}^i \Big( \tilde{\nabla} f_i(x_{t,\ell}^i) - \nabla f_i(x_{t,\ell}^i) + \nabla f_i(x_{t,\ell}^i) \Big) \right\|^2$$

$$\leqslant \frac{4K\tilde{L}^3}{m} \sum_{i=1}^{m} \sum_{k=0}^{K-1} (\eta_{t,k}^i)^2 \left[ \mathbb{E} \left\| \sum_{\ell=0}^{k-1} \eta_{t,l}^i \Big( \tilde{\nabla} f_i(x_{t,\ell}^i) - \nabla f_i(x_{t,\ell}^i) \Big) \right\|^2 + \mathbb{E} \left\| \sum_{\ell=0}^{k-1} \eta_{t,l}^i \nabla f_i(x_{t,\ell}^i) \right\|^2 \right]$$

$$\leqslant \frac{4K\tilde{L}^3}{m} \sum_{i=1}^{m} \sum_{k=0}^{K-1} k(\eta_{t,k}^i)^2 \sum_{\ell=0}^{k-1} (\eta_{t,l}^i)^2 \left[ \mathbb{E} \left\| \tilde{\nabla} f_i(x_{t,\ell}^i) - \nabla f_i(x_{t,\ell}^i) \right\|^2 + \mathbb{E} \left\| \nabla f_i(x_{t,\ell}^i) \right\|^2 \right]$$

$$\leqslant \frac{4K\tilde{L}^3}{m} \sum_{i=1}^{m} \sum_{k=0}^{K-1} k(\eta_{t,k}^i)^2 \sum_{\ell=0}^{k-1} (\eta_{t,l}^i)^2 \left( \frac{\sigma^2}{b} + G^2 \right)$$

$$\leqslant \frac{4K^2\tilde{L}^3 \left( \frac{\sigma^2}{b} + G^2 \right)}{m} \sum_{i=1}^{m} \left( \sum_{k=0}^{K-1} (\eta_{t,k}^i)^2 \right)^2. \tag{25}$$

For $A_5$, using (7), (8), and (12), we have:

$$A_5 \leqslant \frac{2K\tilde{L}}{m} \sum_{i=1}^{m} \sum_{k=0}^{K-1} (\eta_{t,k}^i)^2 \, \mathbb{E} \left\| \nabla f_i(x_t) - \nabla f(x_t) \right\|^2$$

$$\leqslant \frac{2K\tilde{L}\rho}{m} \left\| \nabla f(x_t) \right\|^2 \sum_{i=1}^{m} \sum_{k=0}^{K-1} (\eta_{t,k}^i)^2. \tag{26}$$

For $A_6$, using (7), (8), and (12), we have:

$$A_6 \leqslant \frac{2K\tilde{L}}{m} \left\| \nabla f(x_t) \right\|^2 \sum_{i=1}^{m} \sum_{k=0}^{K-1} (\eta_{t,k}^i)^2. \tag{27}$$

**Putting all bounds together.** Now, by putting (20)–(27) back into (19) and rearranging the terms, we obtain the following inequality:

$$\left( \frac{D_t}{2} - \frac{K\rho}{D_t m} \sum_{i=1}^{m} \sum_{k=0}^{K-1} (\eta_{t,k}^i)^2 - \frac{2K\tilde{L}(1+\rho)}{m} \sum_{i=1}^{m} \sum_{k=0}^{K-1} (\eta_{t,k}^i)^2 \right) \|\nabla f(x_t)\|^2$$

$$\leqslant f(x_t) - f(x_{t+1}) + \frac{2K^2\tilde{L}^2\left( \frac{\sigma^2}{b} + G^2 \right)}{D_t m} \sum_{i=1}^{m} \left( \sum_{k=0}^{K-1} (\eta_{t,k}^i)^2 \right)^2$$

$$+ \frac{2K\tilde{L}\sigma^2}{mb} \sum_{i=1}^{m} \sum_{k=0}^{K-1} (\eta_{t,k}^i)^2 + \frac{4K^2\tilde{L}^3\left( \frac{\sigma^2}{b} + G^2 \right)}{m} \sum_{i=1}^{m} \left( \sum_{k=0}^{K-1} (\eta_{t,k}^i)^2 \right)^2. \tag{28}$$

Now, it remains to notice that, by definition, $\eta_{t,k}^i = \mathcal{O}(\gamma)$, for all $i \in [m]$, $k \in \{0, 1, \ldots K-1\}$, and $t \geqslant 0$. Therefore, $D_t = \mathcal{O}(\gamma K)$. Thus, by dividing the above inequality by $\frac{D_t}{2}$, we have:

$$\left( 1 - \underbrace{\frac{2K\rho}{D_t^2 m} \sum_{i=1}^{m} \sum_{k=0}^{K-1} (\eta_{t,k}^i)^2}_{\mathcal{O}(1)} - \underbrace{\frac{4K\tilde{L}(1+\rho)}{D_t m} \sum_{i=1}^{m} \sum_{k=0}^{K-1} (\eta_{t,k}^i)^2}_{\mathcal{O}(\gamma K)} \right) \|\nabla f(x_t)\|^2$$

$$\leqslant \underbrace{\frac{2(f(x_t) - f(x_{t+1}))}{D_t}}_{\mathcal{O}\left( \frac{1}{\gamma K} \right)} + \underbrace{\frac{4K\tilde{L}\sigma^2}{D_t mb} \sum_{i=1}^{m} \sum_{k=0}^{K-1} (\eta_{t,k}^i)^2}_{\mathcal{O}(\gamma K)}$$

$$+ \underbrace{\frac{4K^2\tilde{L}^2\left( \frac{\sigma^2}{b} + G^2 \right)}{D_t^2 m} \sum_{i=1}^{m} \left( \sum_{k=0}^{K-1} (\eta_{t,k}^i)^2 \right)^2}_{\mathcal{O}(\gamma^2 K^2)}$$

$$+ \underbrace{\frac{8K^2\tilde{L}^3\left( \frac{\sigma^2}{b} + G^2 \right)}{D_t m} \sum_{i=1}^{m} \left( \sum_{k=0}^{K-1} (\eta_{t,k}^i)^2 \right)^2}_{\mathcal{O}(\gamma^3 K^3)}, \tag{29}$$

for $\rho = \mathcal{O}(1)$. For the simplicity of exposition, assuming $\rho \leqslant \frac{1}{4}$ and $T \geqslant (5/\rho)^2 = 400$, we obtain the below by averaging (29) for $t = 0, 1, \ldots, T-1$:

$$\frac{1}{T} \sum_{t=0}^{T-1} \|\nabla f(x_t)\|^2 \leqslant \mathcal{O}\left( \frac{f(x_0) - f^\star}{\gamma K T} \right) + \mathcal{O}\left( \frac{\gamma K \sigma^2}{b} \right) + \mathcal{O}\left( \tilde{L}^2 \left( \frac{\sigma^2}{b} + G^2 \right) \gamma^2 K^2 \right)$$

$$+ \mathcal{O}\left( \tilde{L}^3 \left( \frac{\sigma^2}{b} + G^2 \right) \gamma^3 K^3 \right). \tag{30}$$

Finally, by choosing $\gamma = \mathcal{O}\left(\frac{1}{K\sqrt{T}}\right)$, and defining $\Psi_1 = \max\left\{\frac{\sigma^2}{b}, f(x_0) - f(x^\star)\right\}$ and $\Psi_2 = \left(\frac{\sigma^2}{b} + G^2\right)$ with $b = |\mathcal{B}|$ being the batch size, we arrive at the statement in Theorem 1, i.e.,

$$\frac{1}{T}\sum_{t=0}^{T-1}\mathbb{E}\|\nabla f(x_t)\|^2 \leqslant \mathcal{O}\left(\frac{\Psi_1}{\sqrt{T}}\right) + \mathcal{O}\left(\frac{\tilde{L}^2\Psi_2}{T}\right) + \mathcal{O}\left(\frac{\tilde{L}^3\Psi_2}{\sqrt{T^3}}\right).$$

$\square$

## A.2 PROOF OF ALGORITHM 1 FOR CONVEX CASE

Here, we provide an extension of the proof of Malitsky & Mishchenko (2020, Theorem 1) for the distributed version. Note that the proof we provide here is not exactly the same version of $\Delta$-SGD presented in Algorithm 1; in particular, we assume no local iterations, i.e., $k = 1$ for all $i \in [m]$, and every client participate, i.e, $p = 1$, without stochastic gradients.

We start by constructing a Lyapunov function for the distributed objective in (1).

**Lemma 4.** *Let $f_i : \mathbb{R}^d \to \mathbb{R}$ be convex with locally Lipschitz gradients. Then, the sequence $\{x_t\}$ generated by Algorithm 1, assuming $k = 1$ and $p = 1$ with full batch, satisfy the following:*

$$\|x_{t+1} - x^\star\|^2 + \frac{1}{2m}\sum_{i=1}^{m}\|x_{t+1}^i - x_t^i\|^2 + \frac{2}{m}\sum_{i=1}^{m}\left[\eta_t^i(1 + \theta_t^i)\left(f_i(x_t^i) - f_i(x^\star)\right)\right]$$

$$\leqslant \|x_t - x^\star\|^2 + \frac{1}{2m}\sum_{i=1}^{m}\|x_t^i - x_{t-1}^i\|^2 + \frac{2}{m}\sum_{i=1}^{m}\left[\eta_t^i\theta_t^i\left(f_i(x_{t-1}^i) - f_i(x^\star)\right)\right]. \quad (31)$$

The above lemma, which we prove below, constructs a contracting Lyapunov function, which can be seen from the second condition on the step size $\eta_t^i$ :

$$\eta_{t+1}^i \leqslant \sqrt{1 + \theta_t^i}\eta_t^i \implies \eta_{t+1}^i\theta_{t+1}^i \leqslant (1 + \theta_t^i)\eta_t^i.$$

*Proof.* We denote $x_t = \frac{1}{m}\sum_{i=1}^{m}x_t^i$.

$$\|x_{t+1} - x^\star\|^2 = \|x_t - \frac{1}{m}\sum_{i=1}^{m}\eta_t^i\nabla f_i(x_t^i) - x^\star\| \quad (32)$$

$$= \|x_t - x^\star\|^2 + \left\|\frac{1}{m}\sum_{i=1}^{m}\eta_t^i\nabla f_i(x_t^i)\right\|^2 - 2\left\langle x_t - x^\star, \frac{1}{m}\sum_{i=1}^{m}\eta_t^i\nabla f_i(x_t^i)\right\rangle \quad (33)$$

We will first bound the second term in (33). Using $\|\sum_{i=1}^{m}a_i\|^2 \leqslant m \cdot \sum_{i=1}^{m}\|a_i\|^2$, we have

$$\left\|\frac{1}{m}\sum_{i=1}^{m}\eta_t^i\nabla f_i(x_t^i)\right\|^2 = \left\|\frac{1}{m}\sum_{i=1}^{m}(x_t^i - x_{t+1}^i)\right\|^2 \leqslant \frac{1}{m}\sum_{i=1}^{m}\|x_t^i - x_{t+1}^i\|^2. \quad (34)$$

To bound $\|x_t^i - x_{t+1}^i\|^2$, observe that

$$\begin{aligned}\|x_t^i - x_{t+1}^i\|^2 &= 2\|x_t^i - x_{t+1}^i\|^2 - \|x_t^i - x_{t+1}^i\|^2 \\ &= 2\langle -\eta_t^i\nabla f_i(x_t^i), x_t^i - x_{t+1}^i\rangle - \|x_t^i - x_{t+1}^i\|^2 \\ &= 2\eta_t^i\langle\nabla f_i(x_t^i) - \nabla f_i(x_{t-1}^i), x_{t+1}^i - x_t^i\rangle - \|x_t^i - x_{t+1}^i\|^2 \\ &\quad + 2\eta_t^i\langle\nabla f_i(x_{t-1}^i), x_{t+1}^i - x_t^i\rangle\end{aligned} \quad (35)$$

For the first term in (35), using Cauchy-Schwarz and Young's inequalities as well as $\eta_t^i \leqslant \frac{\|x_t^i - x_{t-1}^i\|}{2\|\nabla f_i(x_t^i) - \nabla f_i(x_{t-1}^i)\|}$, we have:

$$2\eta_t^i\langle\nabla f_i(x_t^i) - \nabla f_i(x_{t-1}^i), x_{t+1}^i - x_t^i\rangle \leqslant 2\eta_t^i\|\nabla f_i(x_t^i) - \nabla f_i(x_{t-1}^i)\| \cdot \|x_{t+1}^i - x_t^i\|$$

$$\leqslant \|x_t^i - x_{t-1}^i\| \cdot \|x_{t+1}^i - x_t^i\|$$
$$\leqslant \frac{1}{2}\|x_t^i - x_{t-1}^i\|^2 + \frac{1}{2}\|x_{t+1}^i - x_t^i\|^2.$$

For the third term in (35), by convexity of $f_i(x)$, we have

$$2\eta_t^i\langle\nabla f_i(x_{t-1}^i), x_{t+1}^i - x_t^i\rangle = 2\frac{\eta_t^i}{\eta_{t-1}^i}\langle x_{t-1}^i - x_t^i, x_{t+1}^i - x_t^i\rangle$$
$$= 2\theta_t^i\eta_t^i\langle x_{t-1}^i - x_t^i, \nabla f_i(x_t^i)\rangle$$
$$\leqslant 2\theta_t^i\eta_t^i[f_i(x_{t-1}^i) - f_i(x_t^i)].$$

Together, we have

$$\frac{1}{m}\sum_{i=1}^m\|x_t^i - x_{t+1}^i\|^2 \leqslant \frac{1}{m}\sum_{i=1}^m\left\{\tfrac{1}{2}\|x_t^i - x_{t-1}^i\|^2 - \tfrac{1}{2}\|x_{t+1}^i - x_t^i\|^2 + 2\theta_t^i\eta_t^i[f_i(x_{t-1}^i) - f_i(x_t^i)]\right\}. \tag{36}$$

Now to bound the last term in (33), we have

$$-2\Big\langle x_t - x^\star, \frac{1}{m}\sum_{i=1}^m\eta_t^i\nabla f_i(x_t^i)\Big\rangle = \frac{2}{m}\eta_t^i\sum_{i=1}^m\Big\langle x^\star - x_t^i, \nabla f_i(x_t^i)\Big\rangle$$
$$\leqslant \frac{2}{m}\sum_{i=1}^m\eta_t^i\big(f_i(x^\star) - f_i(x_t^i)\big), \tag{37}$$

where in the equality we used the fact that $x_t^i = x_t$, for $k = 1$.

Putting (36) and (37) back to (33), we have

$$\|x_{t+1} - x^\star\|^2 = \|x_t - x^\star\|^2 + \Big\|\frac{1}{m}\sum_{i=1}^m\eta_t^i\nabla f_i(x_t^i)\Big\|^2 - 2\Big\langle x_t - x^\star, \frac{1}{m}\sum_{i=1}^m\eta_t^i\nabla f_i(x_t^i)\Big\rangle$$

$$\leqslant \|x_t - x^\star\|^2 + \frac{1}{m}\sum_{i=1}^m\left\{\tfrac{1}{2}\|x_t^i - x_{t-1}^i\|^2 - \tfrac{1}{2}\|x_{t+1}^i - x_t^i\|^2 + 2\theta_t^i\eta_t^i[f_i(x_{t-1}^i) - f(x_t^i)]\right\}$$

$$- 2\Big\langle x_t - x_t^i, \frac{1}{m}\sum_{i=1}^m\eta_t^i\nabla f_i(x_t^i)\Big\rangle + 2\frac{1}{m}\sum_{i=1}^m\eta_t^i\big(f_i(x^\star) - f_i(x_t^i)\big). \tag{38}$$

Averaging (38) for $i = 1, \ldots, m$, notice that the first term in (38) disappears, as $\hat{x}_t = \frac{1}{m}\sum_{i=1}^m x_i$.

$$\|x_{t+1} - x^\star\|^2 + \frac{1}{m}\sum_{i=1}^m\left[2\eta_t^i(1 + \theta_t^i)\big(f_i(x_t^i) - f_i(x^\star)\big) + \frac{1}{2}\|x_{t+1}^i - x_t^i\|^2\right]$$

$$\leqslant \|x_t - x^\star\|^2 + \frac{1}{m}\sum_{i=1}^m\left[2\eta_t^i\theta_t^i\big(f_i(x_{t-1}^i) - f_i(x^\star)\big) + \frac{1}{2}\|x_t^i - x_{t-1}^i\|^2\right].$$

$\square$

**Theorem 5.** *Let $f_i : \mathbb{R}^d \to \mathbb{R}$ be convex with locally Lipschitz gradients. Then, the sequence $\{x_t\}$ generated by Algorithm 1, assuming $k = 1$ and $p = 1$ with full batch, satisfy the following:*

$$\frac{1}{m}\sum_{i=1}^m\big(f_i(\tilde{x}_t^i) - f_i(x^\star)\big) \leqslant \frac{2D\hat{L}^2}{2\hat{L}t + 1}, \tag{39}$$

*where*

$$\tilde{x}_t^i = \frac{\eta_t^i(1 + \theta_t^i)x_t^i + \sum_{k=1}^{t-1}\alpha_k^i x_k^i}{S_t^i},$$
$$\alpha_k^i = \eta_k^i(1 + \theta_k^i) - \eta_{k+1}^i\theta_{k+1}^i$$
$$S_t^i = \eta_t^i(1 + \theta_t^i) + \sum_{k=1}^{t-1}\alpha_k^i = \sum_{k=1}^t\eta_k^i + \eta_1^i\theta_1^i,$$

*and $\hat{L}$ is a constant. Further, if $x^\star$ is any minimum of $f_i(x)$ for all $i \in [m]$, (39) implies convergence for problem (1).*

*Proof.* We start by telescoping (31). Then, we arrive at

$$\|x_{t+1} - x^\star\|^2 + \frac{1}{2m}\sum_{i=1}^m \|x_{t+1}^i - x_t^i\|^2 + \frac{2}{m}\sum_{i=1}^m \left[ \eta_t^i(1+\theta_t^i)\big(f_i(x_t^i) - f_i(x^\star)\big) \right.$$

$$\left. + \sum_{k=1}^{t-1}\big(\eta_k^i(1+\theta_k^i) - \eta_{k+1}^i\theta_{k+1}^i\big)\big(f_i(x_k^i) - f_i(x^\star)\big) \right] \quad (40)$$

$$\leqslant \|x_1 - x^\star\|^2 + \frac{1}{2m}\sum_{i=1}^m \|x_1^i - x_0^i\|^2 + \frac{2}{m}\sum_{i=1}^m \eta_1^i\theta_1^i\big(f_i(x_0^i) - f_i(x^\star)\big) := D. \quad (41)$$

Observe that (40) is nonnegative by the definition of $\eta_t^i$, implying the iterates $\{x_t^i\}$ are bounded. Now, define the set $\mathcal{S} := \mathrm{conv}(\{x_0^i, x_1^i, \dots\})$, which is bounded as the convex hull of bounded points. Therefore, $\nabla f_i$ is Lipschitz continuous on $\mathcal{S}$. That is, there exist $\hat{L}_i$ such that

$$\|\nabla f_i(x) - \nabla f_i(y)\| \leqslant \hat{L}_i\|x - y\|, \quad \forall x, y \in \mathcal{S}.$$

Also note that (40) is of the form $\alpha_t^i[f_i(x_t^i) - f_i(x^\star)] + \sum_{k=1}^{t-1}\alpha_k^i[f_i(x_k^i) - f_i(x^\star)] = \sum_{k=1}^t \alpha_k^i[f_i(x_k^i) - f_i(x^\star)]$. The sum of the coefficients equals:

$$\sum_{k=1}^t \alpha_k^i = \eta_t^i(1+\theta_t^i) + \sum_{k=1}^{t-1}[\eta_k^i(1+\theta_k^i) - \eta_{k+1}^i\theta_{k+1}^i] = \eta_1^i\theta_1^i + \sum_{k=1}^t \eta_k^i := S_t^i. \quad (42)$$

Therefore, by Jensen's inequality, we have

$$D \geqslant \|x_1 - x^\star\|^2 + \frac{1}{2m}\sum_{i=1}^m \|x_1^i - x_0^i\|^2 + 2S_t^i\big(f_i(\tilde{x}_t^i) - f_i(x^\star)\big) \geqslant 2S_t^i\big(f_i(\tilde{x}_t^i) - f_i(x^\star)\big), \quad (43)$$

which implies $f_i(\tilde{x}_t^i) - f_i(x^\star) \leqslant \frac{D^i}{2S_t^i}$, where

$$\tilde{x}_t^i = \frac{\eta_t^i(1+\theta_t^i)x_t^i + \sum_{k=1}^{t-1}\alpha_k^i x_k^i}{S_t^i},$$

$$\alpha_k^i = \eta_k^i(1+\theta_k^i) - \eta_{k+1}^i\theta_{k+1}^i$$

$$S_t^i = \eta_t^i(1+\theta_t^i) + \sum_{k=1}^{t-1}\alpha_k^i = \sum_{k=1}^t \eta_k^i + \eta_1^i\theta_1^i. \quad (44)$$

Now, notice that by definition of $\eta_t^i$, we have

$$\eta_t^i = \frac{\|x_t^i - x_{t-1}^i\|}{2\|\nabla f(x_t^i) - \nabla f(x_{t-1}^i)\|} \geqslant \frac{1}{2\hat{L}_i}, \quad \forall i \in [m].$$

Also, notice that $\eta_1^i\theta_1^i = \frac{(\eta_1^i)^2}{\eta_0^i} = (\eta_1^i)^2$, assuming for simplicity that $\eta_0^i = 1$. Thus, going back to (44), we have

$$S_t^i = \sum_{k=1}^t \eta_k^i + \eta_1^i\theta_1^i \geqslant \sum_{k=1}^t \frac{1}{2\hat{L}_i} + \frac{1}{4\hat{L}_i^2}$$

$$= \frac{t}{2\hat{L}_i} + \frac{1}{4\hat{L}_i^2} \geqslant \frac{t}{2\hat{L}} + \frac{1}{4\hat{L}^2},$$

where $\hat{L} = \max_i \hat{L}_i$. Finally, we have

$$\frac{D}{2} \geqslant \frac{1}{m}\sum_{i=1}^m S_t^i\big(f_i(\tilde{x}_t^i) - f_i(x^\star)\big)$$

$$\geqslant \frac{1}{m}\sum_{i=1}^m \left(\frac{t}{2\hat{L}} + \frac{1}{4\hat{L}^2}\right)\big(f_i(\tilde{x}_t^i) - f_i(x^\star)\big),$$

which implies

$$\frac{2D\hat{L}^2}{2\hat{L}t + 1} \geqslant \frac{1}{m}\sum_{i=1}^m \big(f_i(\tilde{x}_t^i) - f_i(x^\star)\big).$$

$$\square$$

# B   ADDITIONAL EXPERIMENTS AND PLOTS

## B.1   EASE OF TUNING

In this subsection, we show the effect of using different parameter $\delta$ in the step size of $\Delta$-SGD. As mentioned in Section 4, for $\Delta$-SGD, we append $\delta$ in front of the second condition, and hence the step size becomes

$$\eta_{t,k}^i = \min\left\{ \frac{\gamma\|x_{t,k}^i - x_{t,k-1}^i\|}{2\|\tilde{\nabla}f_i(x_{t,k}^i) - \tilde{\nabla}f_i(x_{t,k-1}^i)\|}, \sqrt{1 + \delta\theta_{t,k-1}^i}\,\eta_{t,k-1}^i \right\}.$$

The experimental results presented in Table 1 were obtained using a value of $\delta = 0.1$. For $\gamma$, we remind the readers that we did not change from the default value $\gamma = 2$ in the original implementation in https://github.com/ymalitsky/adaptive_GD/blob/master/pytorch/optimizer.py.

In Figure 4, we display the final accuracy achieved when using different values of $\delta$ from the set $\{0.01, 0.1, 1\}$. Interestingly, we observe that $\delta$ has very little impact on the final accuracy. This suggests that $\Delta$-SGD is remarkably robust and does not require much tuning, while consistently achieving higher test accuracy compared to other methods in the majority of cases as shown in Table 1.

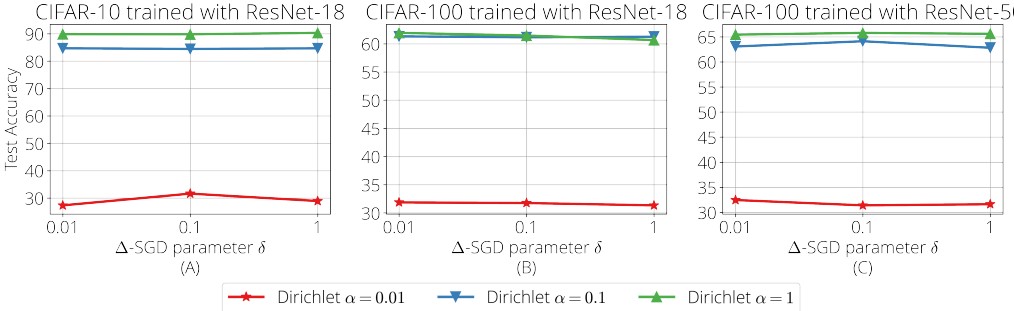

Figure 4: Effect of using different $\delta$ in the second condition of the step size of $\Delta$-SGD.

## B.2   EFFECT OF DIFFERENT NUMBER OF LOCAL ITERATIONS

In this subsection, we show the effect of the different number of local epochs, while fixing the experimental setup to be the classification of CIFAR-100 dataset, trained with a ResNet-50. We show for all three cases of the Dirichlet concentration parameter $\alpha \in \{0.01, 0.1, 1\}$. The result is shown in Figure 5.

As mentioned in the main text, instead of performing local iterations, we instead perform local epochs $E$, similarly to (Reddi et al., 2021; Li et al., 2020). All the results in Table 1 use $E = 1$, and in Figure 5, we show the results for $E \in \{1, 2, 3\}$. Note that in terms of $E$, the considered numbers might be small difference, but in terms of actual local gradient steps, they differ significantly.

As can be seen, using higher $E$ leads to slightly faster convergence in all cases of $\alpha$. However, the speed-up seems marginal compared to the amount of extra computation time it requires (i.e., using $E = 2$ takes roughly twice more total wall-clock time than using $E = 1$). Put differently, $\Delta$-SGD performs well with only using a single local epoch per client ($E = 1$), and still can achieve great performance.

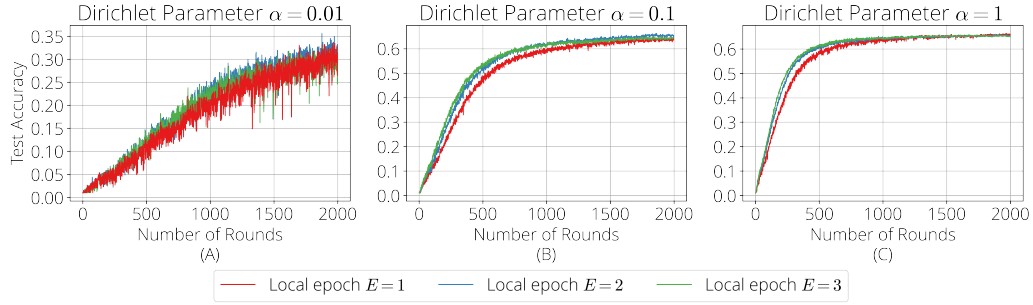

Figure 5: *Effect of the different number of local epochs.*

### B.3 ADDITIONAL EXPERIMENTS USING DIFFERENT NUMBER OF LOCAL DATA PER CLIENT

| *Non-iidness* | *Optimizer* | *Dataset / Model* | |
|---|---|---|---|
| | | CIFAR-10 ResNet-18 | CIFAR-100 ResNet-50 |
| $\mathrm{Dir}(\alpha \cdot \mathbf{p})$ | | | |
| | SGD | $\mathbf{80.7}_{\downarrow(0.0)}$ | $53.5_{\downarrow(4.0)}$ |
| | SGD ($\downarrow$) | $78.8_{\downarrow(1.9)}$ | $53.6_{\downarrow(3.9)}$ |
| | SGDM | $75.0_{\downarrow(5.7)}$ | $53.9_{\downarrow(3.6)}$ |
| | SGDM ($\downarrow$) | $66.6_{\downarrow(14.1)}$ | $53.1_{\downarrow(4.4)}$ |
| $\alpha = 0.1$ | Adam | $79.9_{\downarrow(0.8)}$ | $51.1_{\downarrow(6.4)}$ |
| | Adagrad | $79.3_{\downarrow(1.4)}$ | $44.5_{\downarrow(13.0)}$ |
| | SPS | $64.4_{\downarrow(16.3)}$ | $37.2_{\downarrow(20.3)}$ |
| | $\Delta$-SGD | $\mathbf{80.4}_{\downarrow(0.3)}$ | $\mathbf{57.5}_{\downarrow(0.0)}$ |

Table 3: *Experimental results using different number of local data per client.* Performance difference within 0.5% of the best result for each task are shown in **bold**. Subscripts$_{\downarrow(x.x)}$ is the performance difference from the best result and is highlighted in pink when the difference is bigger than 2%. The symbol ($\downarrow$) appended to SGD and SGDM indicates step-wise learning rate decay, where the step sizes are divided by 10 after 50%, and another by 10 after 75% of the total rounds.

In this subsection, we provide additional experimental results, where the size of the local dataset each client has differs. In particular, instead of distributing 500 samples to each client as done in Section 4 of the main text, we distribute $n_i \in [100, 500]$ samples for each client $i$, where $n_i$ is the number of samples that client $i$ has, and is randomly sampled from the range $[100, 500]$. We keep all other settings the same.

Then, in the server aggregation step (line 13 of Algorithm 1), we compute the weighted average of the local parameters, instead of the plain average, as suggested in (McMahan et al., 2017), and used for instance in (Reddi et al., 2021).

We experiment in two settings: CIFAR-10 dataset classification trained with a ResNet-18 (He et al., 2016), with Dirichlet concentration parameter $\alpha = 0.1$, which was the setting where the algorithms were fined-tuned using grid search; please see details in Section 4. We also provide results for CIFAR-100 dataset classification trained with a ResNet-50, with the same $\alpha$.

The result can be found in Table 3. Similarly to the case with same number of data per client, *i.e.*, $n_i = 500$ for all $i \in [m]$, we see that $\Delta$-SGD achieves good performance without any additional tuning. In particular, for CIFAR-10 classification trained with a ResNet-18, $\Delta$-SGD achieves the second best performance after SGD (without LR decay), with a small margin. Interestingly, SGDM (both with and without LR decay) perform much worse than SGD. This contrasts with the same setting in Table 1, where SGDM performed better than SGD, which again complicates how one should tune client optimizers in the FL setting. For CIFAR-100 classification trained with ResNet-50, $\Delta$-SGD outperforms all other methods with large margin.

## B.4 Additional experiments using FedAdam

As mentioned in Section 2 of the main text, since our work focuses on *client adaptivity*, our method can seamlessly be combined with server-side adaptivity like FedAdam (Reddi et al., 2021). Here, we provide additional experiments using the FedAdam server optimizer.

A big downside of the adaptive server-side approach like FedAdam is that it adds at least one more hyperparameter to tune: the step size for the global optimizer, set aside the two "momentum" parameters ($\beta_1$ and $\beta_2$) in the case of Adam. This necessity to tune additional parameters is quite orthogonal to the approach we take, where we try to *remove the need for step size tuning.*

Indeed, as can be seen in Reddi et al. (2021)[Appendix D.2], 9 different server-side global optimizer step size is grid-searched, and Reddi et al. (2021)[Table 8] shows the best performing server learning rate differs for each task.

That being said, we simply used the default step size and momentum values of Adam (from `Pytorch`), to showcase that our method can seamlessly combined with FedAdam. Due to time constraints, we tested up to CIFAR-100 trained with a ResNet-18, so that all datasets are covered. The result can be found below.

For three out of four cases, $\Delta$-SGD equipped with Adam server-side optimizer achieves the best accuracy with large margin. For the last case, CIFAR-100 trained with a ResNet-18, SGD with decaying step size achieves noticeably better accuracy than $\Delta$-SGD; however, we refer the readers to Table 1 where for the same setting, $\Delta$-SGD equipped with simple averaging achieves 61.1% accuracy, *without any tuning.*

| Additional experiments using FedAdam (Reddi et al., 2021) | | | | | |
|---|---|---|---|---|---|
| *Non-iidness* | *Optimizer* | *Dataset / Model* | | | |
| $\mathrm{Dir}(\alpha \cdot \mathbf{p})$ | | MNIST CNN | FMNIST CNN | CIFAR-10 ResNet-18 | CIFAR-100 ResNet-18 |
| | SGD | $97.3_{\downarrow(0.6)}$ | $83.7_{\downarrow(2.1)}$ | $52.0_{\downarrow(11.8)}$ | $46.7_{\downarrow(2.5)}$ |
| | SGD ($\downarrow$) | $96.4_{\downarrow(1.4)}$ | $80.9_{\downarrow(4.9)}$ | $49.1_{\downarrow(14.7)}$ | $\mathbf{49.2}_{\downarrow(0.0)}$ |
| | SGDM | $\mathbf{97.5}_{\downarrow(0.4)}$ | $84.6_{\downarrow(1.2)}$ | $53.7_{\downarrow(10.1)}$ | $13.3_{\downarrow(35.9)}$ |
| $\alpha = 0.1$ | SGDM ($\downarrow$) | $96.4_{\downarrow(1.5)}$ | $81.8_{\downarrow(4.0)}$ | $53.3_{\downarrow(10.5)}$ | $16.8_{\downarrow(32.4)}$ |
| | Adam | $96.4_{\downarrow(1.5)}$ | $81.5_{\downarrow(4.3)}$ | $27.8_{\downarrow(36.0)}$ | $38.3_{\downarrow(10.9)}$ |
| | Adagrad | $95.7_{\downarrow(2.2)}$ | $82.1_{\downarrow(3.7)}$ | $10.4_{\downarrow(53.4)}$ | $1.0_{\downarrow(48.2)}$ |
| | SPS | $96.6_{\downarrow(1.3)}$ | $85.0_{\downarrow(0.8)}$ | $21.6_{\downarrow(42.2)}$ | $1.6_{\downarrow(47.6)}$ |
| | $\Delta$-SGD | $\mathbf{97.9}_{\downarrow(0.0)}$ | $\mathbf{85.8}_{\downarrow(0.0)}$ | $\mathbf{63.8}_{\downarrow(0.0)}$ | $41.9_{\downarrow(7.3)}$ |

Table 4: *Additional experiments using FedAdam (Reddi et al., 2021).*

## B.5 Additional experiments using MOON

Here, we provide additional experimental results using MOON (Li et al., 2021), which utilizes model-contrastive learning to handle heterogeneous data. In this sense, the goal of MOON is very similar to that of FedProx (Li et al., 2020); yet, due to the model-contrastive learning nature, it can incur bigger memory overhead compared to FedProx, as one needs to keep track of the previous model(s) to compute the representation of each local batch from the local model of last round, as well as the global model.

As such, we only ran using MOON for CIFAR-10 classification using a ResNet-18; for the CIFAR-100 classification using a ResNet-50, our computing resource ran out of memory to run the same configuration we ran in the main text.

The results are summarized in Table 5. As can be seen, even ignoring the fact that MOON utilizes more memory, MOON does not help for most of the optimizers—it actually often results in worse final accuracies than the corresponding results in Table 1 from the main text.

| Non-iidness | Optimizer | Dataset / Model |
|---|---|---|
| $\text{Dir}(\alpha \cdot \mathbf{p})$ | | CIFAR-10
ResNet-18 |
| $\alpha = 0.1$ | SGD | $78.2_{\downarrow(4.9)}$ |
| | SGD ($\downarrow$) | $74.2_{\downarrow(8.9)}$ |
| | SGDM | $76.4_{\downarrow(6.7)}$ |
| | SGDM ($\downarrow$) | $75.5_{\downarrow(14.1)}$ |
| | Adam | $82.4_{\downarrow(0.6)}$ |
| | Adagrad | $81.3_{\downarrow(1.8)}$ |
| | SPS | $9.57_{\downarrow(73.5)}$ |
| | $\Delta$-SGD | $\mathbf{83.1}_{\downarrow(0.0)}$ |

Table 5: *Experimental results using MOON loss function.* Performance differences within 0.5% of the best result for each task are shown in **bold**. Subscripts$_{\downarrow(x.x)}$ is the performance difference from the best result and is highlighted in pink when the difference is bigger than 2%. The symbol ($\downarrow$) appended to SGD and SGDM indicates step-wise learning rate decay, where the step sizes are divided by 10 after 50%, and another by 10 after 75% of the total rounds.

### B.6 ADDITIONAL EXPERIMENTS USING FEDPROX

Here, we provide the complete result of Table 2b, conducting the experiment using FedProx loss function for the most heterogeneous case ($\alpha = 0.01$) for all datasets and architecture considered. The result can be found below, where $\Delta$-SGD achieves the best accuracy in three out of five cases (FMNIST trained with a CNN, CIFAR-10 trained with a ResNet-18, and CIFAR-100 with ResNet-50); in the remaining two cases (MNIST trained with a CNN and CIFAR-100 trained with a ResNet-18), $\Delta$-SGD achieves second-best accuracy with close margin to the best.

| Additional experiments using FedProx loss function (Li et al., 2020) | | | | | | |
|---|---|---|---|---|---|---|
| Non-iidness | Optimizer | Dataset / Model | | | | |
| $\text{Dir}(\alpha \cdot \mathbf{p})$ | | MNIST
CNN | FMNIST
CNN | CIFAR-10
ResNet-18 | CIFAR-100
ResNet-18 | CIFAR-100
ResNet-50 |
| $\alpha = 0.01$ | SGD | $95.7_{\downarrow(1.4)}$ | $79.0_{\downarrow(1.6)}$ | $20.0_{\downarrow(13.8)}$ | $\mathbf{30.3}_{\downarrow(0.0)}$ | $25.2_{\downarrow(5.9)}$ |
| | SGD ($\downarrow$) | $\mathbf{97.2}_{\downarrow(0.0)}$ | $79.3_{\downarrow(1.3)}$ | $31.3_{\downarrow(2.5)}$ | $29.5_{\downarrow(0.8)}$ | $20.2_{\downarrow(10.8)}$ |
| | SGDM | $73.8_{\downarrow(23.4)}$ | $72.8_{\downarrow(7.8)}$ | $29.3_{\downarrow(4.4)}$ | $22.9_{\downarrow(7.5)}$ | $23.8_{\downarrow(7.2)}$ |
| | SGDM ($\downarrow$) | $81.2_{\downarrow(16.0)}$ | $78.0_{\downarrow(2.6)}$ | $25.3_{\downarrow(8.5)}$ | $19.8_{\downarrow(10.5)}$ | $15.0_{\downarrow(16.0)}$ |
| | Adam | $82.3_{\downarrow(14.8)}$ | $65.6_{\downarrow(15.0)}$ | $28.1_{\downarrow(5.7)}$ | $24.8_{\downarrow(5.6)}$ | $22.6_{\downarrow(8.4)}$ |
| | Adagrad | $76.3_{\downarrow(20.9)}$ | $51.2_{\downarrow(29.4)}$ | $19.3_{\downarrow(14.5)}$ | $12.9_{\downarrow(17.4)}$ | $4.1_{\downarrow(26.9)}$ |
| | SPS | $85.5_{\downarrow(11.7)}$ | $62.1_{\downarrow(18.45)}$ | $27.6_{\downarrow(6.2)}$ | $21.3_{\downarrow(9.0)}$ | $16.5_{\downarrow(14.5)}$ |
| | $\Delta$-SGD | $\mathbf{96.9}_{\downarrow(0.26)}$ | $\mathbf{80.6}_{\downarrow(0.0)}$ | $\mathbf{33.8}_{\downarrow(0.0)}$ | $29.7_{\downarrow(0.6)}$ | $\mathbf{31.0}_{\downarrow(0.0)}$ |

Table 6: *Additional experiments using FedProx loss function.*

### B.7 ADDITIONAL PLOT WITH STANDARD DEVIATION (3 RANDOM SEEDS)

In this section, we repeat for three independent trials (using three random seeds) for two tasks: CIFAR-10 classification trained with a ResNet-18, and FMNIST classification trained with a shallow CNN, both with Dirichlet $\alpha = 0.1$,. The aim is to study the robustness of $\Delta$-SGD in comparison with other client optimizers.

We remind the readers that CIFAR-10 classification trained with a ResNet-18 with Dirichlet $\alpha = 0.1$ is where we performed grid-search for each client optimizer, and thus each are using the best step size that we tried. Then, for FMNIST classification, the same step sizes from the previous task are used. The result can be found below.

We see that Δ-SGD not only achieves *the best average test accuracy*, but also achieves *very small standard deviation*. Critically, none of the other client optimizers achieve good performance in *both settings*, other than the exception of Δ-SGD.

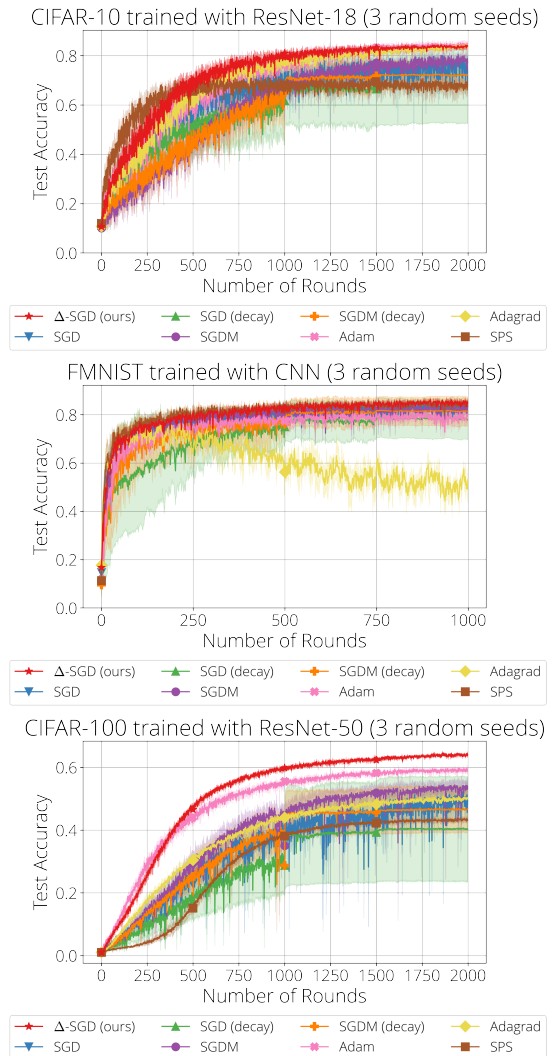

Figure 6: *Additional plots for CIRAR-10 classification trained with a ResNet-18, FMNIST classification trained with a CNN, and CIFAR-100 classification trained with a ResNet-50, repeated for three independent trials.* The average is plotted, with shaded area indicating the standard deviation.

## B.8   STEP SIZE PLOT FOR Δ-SGD

Here, we visualize the step size conditions for Δ-SGD to see how the proposed step size looks like in practice. We only plot the first 300 epochs, as otherwise the plot gets quite messy.

From Figure 7, we can see that both conditions for $\eta_{t,k}^i$ are indeed necessary. The first condition, plotted in green, approximates the local smoothness of $f_i$, but can get quite oscillatory. The second condition, plotted in blue, effectively restricts the first condition from taking too large values.

| Non-iidness | Optimizer | Dataset / Model | | | | | |
|---|---|---|---|---|---|---|---|
| Dir($\alpha \cdot \mathbf{p}$) | | CIFAR-10/ResNet-18 | | FMNIST/CNN | | CIFAR-100/ResNet-50 | |
| | | Average | Std. Dev. | Average | Std. Dev. | Average | Std. Dev. |
| | SGD | $73.96_{\downarrow(9.93)}$ | 0.0263 | $83.43_{\downarrow(1.78)}$ | 0.0178 | $49.44_{\downarrow(14.58)}$ | 0.0376 |
| | SGD ($\downarrow$) | $67.79_{\downarrow(16.10)}$ | 0.1550 | $77.83_{\downarrow(7.38)}$ | 0.0815 | $40.26_{\downarrow(23.76)}$ | 0.1687 |
| | SGDM | $77.59_{\downarrow(6.30)}$ | 0.0142 | $83.27_{\downarrow(1.94)}$ | 0.0186 | $53.80_{\downarrow(10.22)}$ | 0.0149 |
| $\alpha = 0.1$ | SGDM ($\downarrow$) | $72.11_{\downarrow(11.78)}$ | 0.0611 | $81.36_{\downarrow(3.85)}$ | 0.0606 | $46.77_{\downarrow(17.25)}$ | 0.0762 |
| | Adam | $83.24_{\downarrow(0.65)}$ | 0.0126 | $79.98_{\downarrow(5.23)}$ | 0.0228 | $58.89_{\downarrow(5.13)}$ | 0.0098 |
| | Adagrad | $83.86_{\downarrow(0.03)}$ | 0.0048 | $53.06_{\downarrow(32.15)}$ | 0.0487 | $49.93_{\downarrow(14.09)}$ | 0.0177 |
| | SPS | $68.38_{\downarrow(15.51)}$ | 0.0192 | $84.59_{\downarrow(0.62)}$ | 0.0154 | $43.16_{\downarrow(20.86)}$ | 0.0072 |
| | $\Delta$-SGD | $83.89_{\downarrow(0.00)}$ | 0.0052 | $85.21_{\downarrow(0.00)}$ | 0.0152 | $64.02_{\downarrow(0.00)}$ | 0.0051 |

Table 7: *CIRAR-10 classification trained with a ResNet-18 and FMNIST classification trained with a CNN, and CIFAR-100 classification trained with a ResNet-50, repeated for three independent trials. The average among three trials and the standard deviation for each client optimizer are reported.*

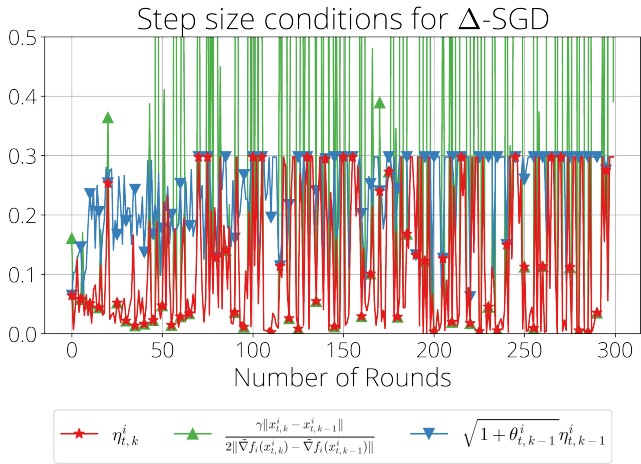

Figure 7: *Step size conditions for $\Delta$-SGD.* In green, we plot the first condition, and in blue, we plot the second condition for $\eta_{t,k}^i$, which is plotted in red color taking the minimum of the two conditions.

### B.9 ILLUSTRATION OF THE DEGREE OF HETEROGENEITY

As mentioned in the main text, the level of "non-iidness" is controlled using latent Dirichlet allocation (LDA) applied to the labels. This approach is based on (Hsu et al., 2019) and involves assigning each client a multinomial distribution over the labels. This distribution determines the labels from which the client's training examples are drawn. Specifically, the local training examples for each client are sampled from a categorical distribution over $N$ classes, parameterized by a vector $\mathbf{q}$. The vector $\mathbf{q}$ is drawn from a Dirichlet distribution with a concentration parameter $\alpha$ times a prior distribution vector $\mathbf{p}$ over $N$ classes.

To vary the level of "non-iidness," different values of $\alpha$ are considered: $0.01, 0.1$, and $1$. A larger $\alpha$ corresponds to settings that are closer to i.i.d. scenarios, while smaller values introduce more diversity and non-iidness among the clients, as visualized in Figure 8. As can be seen, for $\alpha = 1$, each client (each row on $y$-axis) have fairly diverse classes, similarly to the i.i.d. case; as we decrease $\alpha$, the number of classes that each client has access to decreases. With $\alpha = 0.01$, there are man y clients who only have access to a single or a couple classes.



Figure 8: *Illustration of the degree of heterogeneity induced by using different concentration parameter $\alpha$ for Dirichlet distribution, for CIFAR-10 dataset (10 colors) and 100 clients (100 rows on y-axis).*

