# OpenReview forum: "Adaptive Federated Learning with Auto-Tuned Clients"
_ICLR.cc/2024/Conference — ICLR 2024 poster_

### Official Review · Reviewer_Yzoy · 2023-10-29

**Soundness:** 3 good
**Presentation:** 4 excellent
**Contribution:** 2 fair
**Rating:** 6
**Confidence:** 3

**Summary:**

This paper demonstrates an interesting phenomenon where not properly tuning the local step size of the GD algorithm might negatively impact performance of an FL system. To address this, this paper proposes an adaptive scheme for setting local step sizes called $\Delta$-SGD, and provides an analysis of its convergence under mild assumptions. Finally, the paper provides a thorough empirical demonstration of the proposed method, comparing it with other optimizers when used by FedAvg.

**Strengths:**

The paper is clearly written and well motivated.

Theorem 1 and its proof in Appendix A seems to be the main contribution. I have checked and believed that the proofs are sound.

The empirical results presented are generally promising.

**Weaknesses:**

1. Novelty:
- I believe $\Delta$-SGD is algorithmically similar to the Adaptive SGD method in the paper "Adaptive Gradient Descent without Descent" (Malitsky and Mishchenko, 2020). While there are some attempts to cite and discuss this paper, I think the authors should be more upfront about it, as it changes the perceived contribution entirely.
- To the best of my knowledge, Malitsky and Mishchenko did not deal with the FL setting, so I believe theorem 1 and its proof are novel. However, I think the authors should provide a proof sketch to help readers understand the nuance of the proving strategy. What is hard/non-trivial about applying Adaptive SGD in the FL setting?

2. Significance:
- Despite the claim that $\Delta$-SGD is complementary to other FL methods that perform server adaptation this is not demonstrated in the experiments.
- I think Table 2b should be expanded to become one of the main result (conduct over all datasets, rather than as an ablation study on 2 datasets).
-  The gain in performance is not significant enough in many cases. The authors should provide some error bar or standard deviation.

**Questions:**

1.The result presentation is a little bit confusing. The small number on the right of Table 1 is the performance difference from the best result, but what is this "best result" referring to? Best among \alpha = 1.0/ 0.1/ 0.01 ?

2. For all methods, the authors performed grid search on CIFAR-10 and apply it on other test scenarios. Why not performing grid search on each individual setting? Is it because of the expensive cost of hyperparameter tuning?

3. If hyperparameter tuning is the bottleneck (and the motivation), how will this method compare to the method FedEx proposed by the paper Federated Hyperparameter Tuning: Challenges, Baselines, and Connections to Weight-Sharing (Khodak et al., 2021)

---

> ### Author Response · Authors · 2023-11-20
> **Thank you for the review!**
>
> Dear Reviewer Yzoy,
>
> Thank you for your valuable comments and feedback. We are encouraged that you found our paper clearly written and the empirical results promising. We also appreciate you taking the time to check our proof.
>
> $\textcolor{purple}{\textbf{Weaknesses}}$
>
> > **Q: $\Delta$-SGD is algorithmically similar to the Adaptive SGD method in the paper [1]?**
>
> **A:** You are right. $\Delta$-SGD is indeed inspired by [1]. We will make the connection more explicit when we introduce $\Delta$-SGD. However, as we detail in the following answer, extending the proof to a distributed nonconvex setting is nontrivial. Further, any empirical studies on how [1] can be developed and how it works in FL settings were missing from the literature; this is the purpose of this work, among other goals.
>
> ---
>
>
> > **Q: [1] did not deal with the FL setting, so Theorem 1 and its proof is novel. But proof sketch would help. What is hard/nontrivial about applying Adaptive SGD in the FL setting?**
>
> **A:** We are glad you asked this question so we can explain in detail the nontriviality of extending proof in [1] to the FL setting. Before delving into this, we first kindly remind you that Theorem 1 is not our only theoretical extension of [1]:
> - As you are aware, in Appendix A.1, we provide the proof of Theorem 1, which is for distributed nonconvex settings and includes stochasticity and local iterations.
> - Appendix A.2 provides proof that extends [1] to a distributed setting where $f_i$ is convex.
> - Appendix A.3 provides a simple proof in the centralized nonconvex setting.
>
> That being said, there are several challenges in extending the theoretical result in [1] to the FL setting, as we briefly wrote before Assumption 1 in Section 3.1. A primary obstacle is that since $\eta_{t, k}^i$ is prescribed using known quantities, we cannot simply *choose* a *common constant step size* to work for all clients. Our proof has to deal with adaptive step size, which is different across all clients. Moreover, we have to account for the "model drift" between the server and the client model parameters due to local iterations performed by each client, which is trickier to handle on top of stochasticity and the individual adaptive step size we use.
>
> We can add this explanation before introducing Assumption 1 in the revision.
>
> ---
>
>
> > **Q: Despite the claim that $\Delta$-SGD is complementary to other FL methods that perform server adaptation, this is not demonstrated in the experiments.**
>
> **A:** Thank you for the comment. *We added FedAdam experiments in Appendix B.1* of the updated supplementary material.
>
> ---
>
>
> > **Q: Table 2b should be expanded to become one of the main result (conduct over all datasets, rather than as an ablation study on 2 datasets).**
>
> **A:** Thank you for the suggestion. *We have added a new table in Appendix B.2* where we have conducted experiments using the FedProx loss function over all datasets, per the suggestion.
>
> ---
>
>
> > **Q:​​ The authors should provide some error bar or standard deviation.**
>
> **A:** Thank you for the suggestion! *We have included additional results in Appendix B.3* of the updated supplementary material, plotting and reporting the average and standard deviation of three independent trials for CIFAR-10 classification trained with a ResNet-18.

---

> ### Author Response · Authors · 2023-11-20
> **Thank you for the review!**
>
> $\textcolor{purple}{\textbf{Questions}}$
>
> > **Q1: The small number on the right of Table 1 is the performance difference from the best result, but what is this "best result" referring to? Best among \alpha = 1.0/ 0.1/ 0.01 ?**
>
> **A:** *Best result* refers to the highest accuracy within a fixed experiment (including a fixed $\alpha$) among *different client optimizers*. For example, for CIFAR-10 classification trained with a ResNet-18 with $\alpha=1$, the best result is achieved by $\Delta$-SGD with the achieved accuracy 89.8. For this setting, SGD achieves 87.7$_{\downarrow(2.1)}$, and $89.8 = 87.7 + 2.1$. We remind the reviewer that $\alpha$ changes client data heterogeneity (c.f., Figure 7 in Appendix B.6), so different $\alpha$ constitute a different experimental setup.
>
> ---
>
>
>
> > **Q2: For all methods, the authors performed grid search on CIFAR-10 and apply it on other test scenarios. Why not performing grid search on each individual setting? Is it because of the expensive cost of hyperparameter tuning?**
>
> **A:** This is precisely why we are proposing $\Delta$-SGD, as it can perform well ***without any additional tuning across different scenarios***. To provide a bit more perspective on how much computing/time effort is needed for step size tuning, let us focus on CIFAR-10 classification trained with a ResNet-18, with a fixed $\alpha$. For this task, SGD takes about 22 seconds per communication round. Since we run 2000 epochs, this amounts to 44,000 seconds, roughly 12 hours. Since we tried four different step sizes, it takes 48 hours *just for SGD*. Now consider adding learning rate decay, on top of that adding momentum, and consider performing grid-search of other client optimizers, not to mention different $\alpha$'s and larger-scale experiments using CIFAR-100 and ResNet-50.
>
> ---
>
>
> > **Q3: How does $\Delta$-SGD compare to the method FedEx proposed by the paper “Federated Hyperparameter Tuning: Challenges, Baselines, and Connections to Weight-Sharing” (Khodak et al., 2021)?**
>
> **A:** Thank you for the reference. While FedEx and $\Delta$-SGD can be loosely connected in the sense of hyperparameter tuning, we clarify that FedEx *requires solving yet another optimization problem (Eq. (6) in [2])*. It is unclear whether solving that optimization problem is computationally more accessible or more challenging than performing a grid search. Our proposed method, $\Delta$-SGD, is fundamentally different because it removes the need for client step size tuning while achieving outstanding performance.
>
> ---
>
>
> We hope the above answers clarify your concerns, and if so, we hope you could consider reflecting on the score. Thank you again for your time in reviewing our manuscript.
>
> **References**
>
> [1] Malitsky and Mishchenko (2020) "Adaptive Gradient Descent without Descent"
>
> [2] Khodak et al. (2021) “Federated Hyperparameter Tuning: Challenges, Baselines, and Connections to Weight-Sharing”

---

> ### Comment · Reviewer_Yzoy · 2023-11-21
> **Discussion**
>
> Thanks for replying to my questions.
>
> - Regarding the claim that "it can perform well without any additional tuning across different scenarios" -- is this true when there are still hyperparameters in your framework (hyper-hyperparameter in this case) such as \gamma, \theta_0 and \eta_0?
>
> - The extra experiment with error bar did more to unconvince me than to convince, seeing that \Delta-SGD, Adam and Adagrad practically converge to the same performance.
>
> - The cost of grid search is quite interesting. It would be insightful to provide a heatmap of model performance on the grid of hyperparameters vs. \Delta-SGD performance, which would demonstrate how likely can other optimizers match the performance of \Delta-SGD if we simply choose a random set of hyperparameters (this is only a suggestion for future work -- I understand it would be hard to perform this exp now).
>
> - Regarding FedEx, I am aware of the inner optimization problem, and I am under the impression that it would be cheaper than doing a dense grid search. That being said, it would be more of an apples-to-apples comparison than your current experiments, seeing that current baselines do not auto-tune hyperparameters at all.

---

> ### Author Response · Authors · 2023-11-21
> **Discussion continued**
>
> Thank you for the additional comments.
>
> > **Q:** Regarding the claim that "it can perform well without any additional tuning across different scenarios" -- is this true when there are still hyperparameters in your framework (hyper-hyperparameter in this case) such as \gamma, \theta_0 and \eta_0?
>
> **A:** Yes, the fact that $\Delta$-SGD can perform well without any additional tuning is the main point we are trying to convey in this paper. As we mention in Section 4 (footnote 3, pg. 6), we use $\gamma = 2$, $\eta_0 = 0.2$, and $\theta_0=1$ for all experiments. From this setting, *without any additional tuning*, $\Delta$-SGD achieves the performance summarized in Table 1: *TOP-1 test accuracy in 73%, and TOP-2 accuracy in 100% of the experiments (in Table 1).*
>
> We also kindly remind you that among 9 additional experiments we added in Appendix B.1 (FedAdam) and B.2 (FedProx) to address your original comments on the weaknesses, $\Delta$-SGD achieves TOP-1 test accuracy in 6/9 settings and TOP-2 test accuracy in 8/9 settings, without any additional tuning.
>
> ---
>
>
> > **Q:** The extra experiment with error bar did more to unconvince me than to convince, seeing that \Delta-SGD, Adam and Adagrad practically converge to the same performance.
>
> **A:** The main point we are trying to emphasize is that *$\Delta$-SGD performs well without any additional tuning across different settings.* We have updated the supplementary material again, where we also added the average and standard deviation for three independent trials of FMNIST classification trained with a shallow CNN (Appendix B.3, Figure 4 and Table 5). For readability, we report the average test accuracies (among three trials) of two tasks below:
>
> |              | CIFAR-10 + ResNet-18 | FMNIST + CNN |
> |:--------------:|:----------:|:--------:|
> | SGD          | 73.96    | 83.43  |
> | SGD decay    | 67.79    | 77.83  |
> | SGDM         | 77.59    | 83.27  |
> | SGDM decay   | 72.11    | 81.36  |
> | Adam         | 83.24    | 79.98  |
> | Adagrad      | 83.86    | 53.06  |
> | SPS          | 68.38    | 84.59  |
> | $\Delta$-SGD | 83.89    | 85.21  |
>
> For FMNIST classification, for instance, Adam and Adagrad perform quite poorly. Critically, **none of the other client optimizers achieve good performance in _both settings_.** $\Delta$-SGD, on the other hand, not only achieves the best average test accuracies in both settings, but also achieves extremely small (either the smallest or the second smallest) standard deviations.
>
> ---
>
>
> > **Q:** The cost of grid search is quite interesting. It would be insightful to provide a heatmap of model performance on the grid of hyperparameters vs. \Delta-SGD performance, which would demonstrate how likely can other optimizers match the performance of \Delta-SGD if we simply choose a random set of hyperparameters (this is only a suggestion for future work -- I understand it would be hard to perform this exp now).
>
> **A:** Thank you for the suggestion! As you noted, we do not have time to include such plot during the discussion period, but we will be happy to include one for the final version.
>
> ---
>
>
> > **Q:** Regarding FedEx, I am aware of the inner optimization problem, and I am under the impression that it would be cheaper than doing a dense grid search. That being said, it would be more of an apples-to-apples comparison than your current experiments, seeing that current baselines do not auto-tune hyperparameters at all.
>
> **A:** *The fact that current baselines are not able to auto-tune hyperparameters at all is exactly why we are proposing $\Delta$-SGD*: it simply can auto-tune itself, and work well across many different settings. We will be happy to try FedEx with other client optimizers, but we want to clarify once again that *removing the need for expensive grid-search is a major point of our proposal, $\Delta$-SGD*.

---

> > ### Comment · Reviewer_Yzoy · 2023-11-23
> > **Thanks for your responses**
> >
> > I appreciate the explanations and the new results. Although I will keep my score for now (which already reflects my positive sentiment), I will consider raising it after my discussion with other reviewers to see if their concerns are addressed.
> >
> > Best,
> > Reviewer Yzoy

---

> > > ### Author Response · Authors · 2023-11-23
> > >
> > > We truly appreciate your feedback and engagement throughout the discussion! Thank you again for taking the time to carefully review our paper and provide constructive feedback.

---

### Official Review · Reviewer_RQMs · 2023-10-30

**Soundness:** 3 good
**Presentation:** 2 fair
**Contribution:** 2 fair
**Rating:** 8
**Confidence:** 3

**Summary:**

The authors proposed using a locality adaptive step size rule for client-side training in Federated Learning frameworks. Their method requires almost no hyperparameter tuning. They showed the superiority of their algorithm by comparing it with other adaptive client-side step-size methods experimentally. They inspired this technique from a similar technique proposed for centralized training. They proved its convergence guarantees for both convex and nonconvex functions.

**Strengths:**

1.	Federated learning frameworks need a careful design to handle the heterogeneity across clients. A local learning rate best for a client may be suboptimal for another. The authors successfully apply a similar technique proposed for centralized ML to the federated problem. Also, they prove the convergence.
2.	Their method allows them to use a milder “L-smooth” assumption than the one commonly used in nonconvex optimization analyses. They use the maximum of “average local smoothness over local steps” across clients and rounds.

**Weaknesses:**

1.	When you presented the results based on the learning rate tuned on one model/dataset and used for all, the baseline methods were not as successful as yours. One natural question is how good those methods are if we tune for each model/dataset separately. Are all methods comparable? Or, how much the not optimum learning rates of baselines are different from the ones optimal for each dataset?
2.	The bounded gradient assumption can be seen as a strong assumption, and it doesn’t hold for many common functions.  Can those terms be bounded using the other two assumptions, bounded variance and dissimilarity?

**Questions:**

1.	In (4), client $i$ seems to use exact gradients to choose the step size. Shouldn’t it be $\tilde{\nabla}{f_i(x_t^i)}- \tilde{\nabla}{f_i(x_{t-1}^i)}$, or how does client $i$ know true gradient? I guess, it is just a notation thing because there doesn’t seem any problem in Algorithm 1.
2.	I couldn’t get the intuition behind setting the initial step $\eta_{t,0}^i$ to an arbitrary $\eta_0$ at the beginning of each local training after the first one (line 6 in Algorithm 1). Wouldn’t it be good to use the final step size of each client's latest local training?
3.	What parameter(s) did you tune for $\Delta-SGD$ except $\gamma$?
4.	“This is a counter-intuitive behavior, as one would expect to get better accuracy by using a more powerful model.” $\rightarrow$ Here, mentioning Adagrad, Adam, and SPS is finetuned on ResNet18 can be useful. I guess, it should be one of the reasons. Also, it would highlight your method doesn’t need any additional tuning.
5.	In (B) and (C) of Figure 3, have the parameters been tuned on MNIST+CNN and CIFAR-10+ResNet-18, respectively?
6.	You may enrich the study by reporting how the learning rate changes across time and across clients in $\Delta-SGD$.  Also, you can compare this with the best learning rate of the other methods.
7.	The two parts of local learning rate calculation (line 9 in Algorithm 1) are clear in how they are used in the theoretical analysis. It may be good to include an experimental report to see in how many iterations the learning rate is updated based on the first part $\left(\frac{\gamma \lVert x^i_{t, k} - x^i_{t, k-1} \rVert}{2 \lVert \tilde{\nabla} f_i(x^i_{t, k}) - \tilde{\nabla} f_i(x^i_{t, k-1}) \rVert}\right)$ or the second part $\left(\sqrt{1 + \theta^i_{t,k-1}} \eta^i_{t,k-1}\right)$. It may show that the minimum of those two is useful in practice as well.

---

> ### Author Response · Authors · 2023-11-20
> **Thank you for the review!**
>
> Dear Reviewer RQMs,
>
> Thank you for your detailed review. We are glad you acknowledge the superiority of $\Delta$-SGD without almost no hyperparameter tuning. Below, we answer your comments and questions in detail.
>
> $\textcolor{purple}{\textbf{Weaknesses}}$
>
> > **Q: How good those methods are if we tune for each model/dataset separately? Also, how much the not optimum learning rates of baselines are different from the ones optimal for each dataset?**
>
> **A:** Thank you for the question. As one can infer from Figure 1 (left panel), all methods perform reasonably well when we tune the step size for each client optimizer. However, we are precisely trying to *remove the need for additional tuning when one changes either the dataset or the model architecture*. We also kindly refer the reviewer to the updated supplementary Appendix B.3, where we added a plot and a table reporting the average and the standard deviation among three independent trials of each client optimizer using the best respective hand-tuned step size, for the task of CIFAR-10 classification trained with a ResNet-18.
>
> Regarding how much the optimal learning rates of baselines differ from those optimal for each dataset, we kindly refer the reviewer to [1], where exactly that information is already present. Table 8 (Appendix D.4, page 29) of [1] shows the best learning rate for different datasets, both for the server and the client. Quite intuitively, the best-performing step size for each dataset differs significantly for each task.
>
> ---
>
>
> > **Q: Bounded gradient assumption can strong. Can it be bounded using the other two assumptions, bounded variance and dissimilarity?**
>
> **A:** In general, gradient norm cannot be bounded with bounded variance and function dissimilarity because bounded variance measures the departure of the stochastic gradients $\tilde{\nabla} f_i(\cdot)$ from their deterministic mean $\nabla f_i(\cdot)$, and the function dissimilarity measures the difference in gradients of the local cost functions $\nabla f_i(\cdot)$ and the average cost function $\nabla f(\cdot)$. We also kindly remind the reviewer that bounded gradient is quite a common assumption in non-convex optimization [2, 3, 4, 5], especially when using an adaptive optimizer. Still, removing the bounded gradient assumption can be an interesting future work to improve the theoretical analysis of our proposed method.

---

> ### Author Response · Authors · 2023-11-20
> **Thank you for the review!**
>
> $\textcolor{purple}{\textbf{Questions}}$
>
> > **Q1: In (4), client $i$ seems to use exact gradients to choose the step size. How does client know true gradient? I guess, it is just a notation thing because there doesn’t seem any problem in Algorithm 1.”**
>
> **A:** Yes, you are right: we used the exact gradients in Eq. (4) for more explicit exposition with simpler notations. As we write right below Eq. (5) (now highlighted in blue color in the updated supplementary material), “For the FL settings, we extend (4) by including stochasticity and local iterations, as summarized in Algorithm 1.”
>
> ---
>
>
> > **Q2: What is the intuition behind setting the initial step $\eta_{t, 0}^i$ to an arbitrary $\eta_0$ at the beginning of each local training after the one (line 6 in Algorithm 1). Wouldn’t it be good to use the final step size of each client’s latest local training?**
>
> **A:** You are correct that using the final step size of each client’s latest local training would be good. We used an arbitrary $\eta_0$ due to how we implemented the federated learning experiments. Regardless, the reason why arbitrary $\eta_0$ is fine is that, with some reasonable $\eta_0$ (e.g., 0.01), $x_{t, 1}^i$ will be reasonably close to $x_{t, 0}^i$. Then $\eta_{t, 1}^i$ is already a good estimate of the (local) smoothness of $f_i$, only with one local iteration (from $k=0$ to $k=1$).
>
> ---
>
>
> > **Q3: What parameter(s) did you tune for $\Delta$-SGD except $\gamma$?**
>
> **A:** *We want to emphasize that we did not tune $\gamma$. As we write right below the pseudocode in pg. 5, $\gamma$ is only needed for Theorem 1.* In practice, we keep the default value $\gamma=2$ from the original implementation; tuning for $\gamma$ might result in even better performance. For other parameters, as we wrote in the paragraph starting with **Hyperparameters** in pg. 6 of the original submission, we append $\delta$ in front of the second condition of $\eta_{t,k}^i$. Still, we use $\delta=0.1$ for all experiments. We also kindly refer the reviewer to Appendix B.3, where we compare the performance of $\Delta$-SGD for different values of $\delta$ to demonstrate the unnecessity of tuning for $\Delta$-SGD.
>
> ---
>
>
> > **Q4: “This is a counter-intuitive behavior, as one would expect to get better accuracy by using a more powerful model.” Here, mentioning Adagrad, Adam, and SPS is finetuned on ResNet18 can be useful. I guess, it should be one of the reasons. Also, it would highlight your method doesn’t need any additional tuning.**
>
> **A:** Thank you! We took your suggestion and modified the supplementary material accordingly (highlighted in blue on pg. 8).
>
> ---
>
>
> > **Q5: In (B) and (C) of Figure 3, have the parameters been tuned on MNIST+CNN and CIFAR-10+ResNet-18, respectively?**
>
> **A:** No, the parameters are tuned using Figure 1 (A). Then, as the caption reads, “The same step size from (A) is intentionally used in settings (B) and (C).” Similar to our main results in Table 1, the purpose of Figure 1 was to highlight two points:
> $i)$ $\Delta$-SGD works well without any tuning across different datasets, model architectures, and degrees of heterogeneity;
> $ii)$ other optimizers perform suboptimally without additional tuning.

---

> ### Author Response · Authors · 2023-11-20
> **Thank you for the review!**
>
> > **Q6: You may enrich the study by reporting how the learning rate changes across time and across clients in $\Delta$-SGD. Also, you can compare this with the best learning rate of the other methods.**
>
> **A:** Thank you for the great suggestion! *We plotted the learning rate conditions of $\Delta$-SGD in Figure 5 of Appendix B.4 of the updated supplementary material*. Our proposed step size adapts to the local smoothness and oscillates between the values as large as 0.3 and as small as $\sim$ 0.0002.
>
> ---
>
>
> > **Q7: The two parts of local learning rate calculation (line 9 in Algorithm 1) are clear in how they are used in the theoretical analysis. It may be good to include an experimental report to see in how many iterations the learning rate is updated based on the first part $\left( \frac{\gamma \|x_{t,k}^i - x_{t,{k-1}}^i \|}{2 \| \tilde{\nabla} f_i(x_{t,k}^i) - \tilde{\nabla} f_i (x_{t,{k-1}}^i)  \|}  \right)$ or the second part $\left(
> \sqrt{ 1 + \theta_{t,k-1}^i } \eta_{t,k-1}^i \right)$. It may show that the minimum of those two is useful in practice as well.**
>
> **A:** Again, we appreciate your great suggestion. We took your recommendation and *plotted the first and second parts and their minimum in Figure 5 of Appendix B.4 in the updated supplementary material.*
>
> From that figure, we can see that *both conditions for $\eta_{t,k}^i$ are necessary.* The first condition, plotted in green, approximates the local smoothness of $f_i,$ but can get quite oscillatory. The second condition, plotted in blue, effectively restricts the first condition from taking too large values. We believe this enriches our paper, as you suggested.
>
> ---
>
>
> We hope the above answers clarify your concerns, and if so, we hope you could consider reflecting on the score. Thank you again for your time in reviewing our manuscript.
>
> **References**
>
> [1] Reddi, et al., (2021) "Adaptive Federated Optimization"
>
> [2] Ward, et al. (2019)  “AdaGrad Stepsizes: Sharp Convergence Over Nonconvex Landscapes”
>
> [3] Koloskova, et al. (2022) "Sharper convergence guarantees for asynchronous sgd for distributed and federated learning." Advances in Neural Information Processing Systems 35
>
> [4] Fallah, et al. (2020) "Personalized federated learning with theoretical guarantees: A model-agnostic meta-learning approach." Advances in Neural Information Processing Systems 33
>
> [5] Nguyen, John, et al. (2022) "Federated learning with buffered asynchronous aggregation." International Conference on Artificial Intelligence and Statistics.

---

> > ### Comment · Reviewer_RQMs · 2023-11-23
> > **Thanks for the answers!**
> >
> > I thank the authors for their detailed explanations and new experiments. They have resolved all of my concerns!
> >
> > I also have closely followed the discussions with other reviewers. I have already found that the paper is above the threshold for the conference. I am considering raising my score after the discussion period with other reviewers to see if their concerns are mostly resolved.

---

> > > ### Author Response · Authors · 2023-11-23
> > >
> > > We truly appreciate your feedback and engagement throughout the discussion! Thank you again for taking the time to carefully review our paper and provide constructive feedback.

---

### Official Review · Reviewer_WWBM · 2023-10-30

**Soundness:** 2 fair
**Presentation:** 3 good
**Contribution:** 3 good
**Rating:** 6
**Confidence:** 3

**Summary:**

The authors provide a new learning rate schedule in federated learning scenarios. Each client can adjust their own learning rates by their own local gradients. The authors show the convergence analysis of the proposed algorithm in the non-convex case. The experiments show the benefit of the proposed algorithm.

**Strengths:**

1. The authors introduce a new learning rate schedule for each local client in the federated learning. Meanwhile, they show the convergence under mild assumptions in the nonconvex case.

2. The experiments show that the proposed algorithm is more robust than other algorithms w.r.t. non-i.i.dness in federated learning.

**Weaknesses:**

The proof does not seem to be correct to me.

***For the proof of Lemma 2:***

(i)  $\eta_t^{i}$ is used for updating from $x_t$ to $x_{t+1}$, but in the proof it seems like the $\eta_t^{i}$ is used for updating from $x_{t-1}$ to $x_t$. If the latter case is necessary for the proof, how can we get $x_t$ and $\nabla f(x_t)$ without knowing $\eta_t^{i}$.

(ii) In the algorithm $\eta_x^{i}$ is based on the gradient of minibatch estimation, how to change them into the true gradient.

***For the proof in A.1:***

Inequality from (12) to (13) does not seem to be correct. Because $\eta_{t,k}^i$ depends on $\tilde{\nabla} f_i (x_{t,k}^t)$, we need to show that the expected value of $\eta_{t,k}^i \tilde{\nabla} f_i (x_{t,k}^t)$. Since they are not independent, it is trivial to see that $E [\eta_{t,k}^i \tilde{\nabla} f_i (x_{t,k}^t]) = E [\eta_{t,k}^i E[\tilde{\nabla} f_i (x_{t,k}^t)]]$

**Questions:**

Can you provide some detailed verification of the proof in the appendix?

---

> ### Author Response · Authors · 2023-11-20
> **Thank you for the review!**
>
> Dear reviewer WWBM,
>
> We thank you for the comments and questions. There seems to be some confusion about our proof, which we clarify below.
>
> ### ***For the proof of Lemma 2***
>
>
> > **Q: $\eta_t^i$ is used for updating from $x_t$ to $x_{t+1}$, but in the proof it seems like the $\eta_t^i$ is used for updating from $x_{t-1}$ to $x_t$. If the latter case is necessary for the proof, how can we get $x_t$ and $\nabla f(x_t)$ without knowing $\eta_t^i$.**
>
> **A**: For simplicity, let us explain with deterministic gradients without local updates in this response. (We will answer about stochasticity in $(ii)$ below.)
>
> It is true that in the algorithm $\eta_t^i$ is used to update $x_{t+1},$ as $x_{t+1}^i = x_t^i - \eta_t^i \nabla f_i (x_t^i).$
>
> **In the proof, we are not *updating* $x_t^i$ *from* $x_{t-1}^i$ *using* $\eta_t^i$, but just *utilizing the definition* of $\eta_t^i$, as we clarify below.**
>
> $\eta_t^i$ is defined as:
> \begin{align}
>     \eta_t^i = \min ( \tfrac{||x_t^i - x_{t-1}^i ||}{2 || \nabla f_i(x_t^i) - \nabla f_i (x_{t-1}^i)  ||},  \sqrt{ 1 + \theta_{t-1}^i } \eta_{t-1}^i ) , \quad \theta_{t-1}^i = \eta_{t-1}^i/\eta_{t-2}^i.
> \end{align}
> Based on the definition above, we have
> \begin{align}
>     \eta_t^i \leq \tfrac{||x_t^i - x_{t-1}^i ||}{2 || \nabla f_i(x_t^i) - \nabla f_i (x_{t-1}^i)  ||} \implies
>     || \nabla f_i(x_t^i) - \nabla f_i(x_{t-1}^i) || \leq \frac{1}{2\eta_t^i} ||x_t^i - x_{t-1}^i ||.
> \end{align}
>
> We hope the above clarifies the original confusion. That said, we slightly modified the statement and the proof of Lemma 2 in the updated supplementary material, where the source of the initial confusion is removed.
>
> ---
>
>
> > **Q: In the algorithm $\eta_x^i$ is based on the gradient of minibatch estimation, how to change them into the true gradient.**
>
> **A:** We are confused about what you mean by "change to true gradient". We assume by $\eta_x^i$, you mean $\eta_{t,k}^i$ used in Algorithm 1. In the updated supplementary material, we clarified the proof of Lemma 2 for readability, which we also explain below.
>
> Since by Assumptions (1a) (bounded variance) and (1b) (bounded gradient) as well as the independence of samples $z\in\mathcal{B}$, we have that stochastic gradients are bounded:
> \begin{align*}
>     \mathbb{E} || \tilde{\nabla}f_i(x) ||^2 = \mathbb{E} || \frac{1}{|\mathcal{B}|} \sum_{z\in\mathcal{B}} \nabla F_i(x,z)  ||^2 \leq \mathbb{E} || \frac{1}{|\mathcal{B}|} \sum_{z\in\mathcal{B}} \nabla F_i(x,z) - \nabla f_i(x)  ||^2 + || \nabla f_i(x) ||^2.
> \end{align*}
>
> Therefore, as we explain at the beginning of the proof of Lemma 2, Eq. (9) holds in the presence of stochastic gradients, and thus Eq. (12) holds.
>
> ---
>
>
> ### ***For the proof in A.1:***
>
> > **Q: Inequality from (12) to (13) does not seem to be correct. Because $\eta_{t,k}^i$ depends on $\tilde{\nabla} f_i(x_{t,k}^i)$, we need to show that the expected value of $\eta_{t, k}^i \tilde{\nabla} f_i(x_{t, k}^i)$. Since they are not independent, it is trivial to see that $E[ \eta_{t, k}^i \tilde{\nabla} f_i(x_{t,k}^i) ] = E[ \eta_{t, k}^i E [ \tilde{\nabla} f_i (x_{t,k}^i) ] ]$**
>
> **A:** We appreciate your question. As stated at the end of page 4, we use the notation
> \begin{align*}
>     \tilde{\nabla} f_i(x) = \frac{1}{|\mathcal{B}|} \sum_{z\in\mathcal{B}} \nabla F_i(x,z)
> \end{align*}
> as a shorthand for the stochastic gradients with batch $\mathcal{B}$. When this term appears in the algorithm/proof, we sample a new independent batch with respect to data distribution $\mathcal{D}_i$.
>
> Therefore, the stepsize and stochastic gradient in line (12) of the proof in A.1 are independent:
>
> \begin{align*}
>     E_{z \sim D_i} [\eta_{t,k}^i \tilde{\nabla}f_i(x_{t,k}^i)] &=
>     \eta_{t,k}^i E_{z \sim D_i} [\tilde{\nabla}f_i(x_{t,k}^i)] ,
> \end{align*}
>
> as $\eta_{t,k}^i$ does not depend on $z,$ but on another independently sampled minibatch.
> This is a well-known proof technique that is commonly used in the following works:
> - Malitsky, et al (2020). "Adaptive gradient descent without descent." 37th International Conference on Machine Learning
> - Fallah, et al (2020). "Personalized federated learning with theoretical guarantees: A model-agnostic meta-learning approach." Advances in Neural Information Processing Systems 33
> - Fallah, et al (2020). "On the convergence theory of gradient-based model-agnostic meta-learning algorithms." International Conference on Artificial Intelligence and Statistics.
> - Kayaalp, et al (2022). "Dif-MAML: Decentralized multi-agent meta-learning." IEEE Open Journal of Signal Processing 3 (2022): 71-93.
>
> We clarified this part in the updated supplementary material (in blue color).
>
> We hope the above answers clarify your concerns, and if so, we hope you could consider reflecting on the score. Thank you again for your time in reviewing our manuscript, and please don't hesitate to follow up if anything is unclear.

---

> > ### Comment · Reviewer_WWBM · 2023-11-22
> >
> > I'm sorry for not making the second question clear.  In the deterministic version, we can get that $\eta_{t,k}^{i} \leq \frac{\gamma ||x_{t,k}^i - x_{t,k-1}^i||}{2 || \tilde{\nabla} f_i(x_{t,k}^i) - \tilde{\nabla} f_i(x_{t,k-1}^i)||}$.
> > However, it seems that what we need is that
> > \\[
> > E(\eta_{t,k}^{i}) \leq E(\frac{\gamma ||x_{t,k}^i - x_{t,k-1}^i||}{2 ||\nabla f_i(x_{t,k}^i) - \nabla f_i(x_{t,k-1}^i)||})
> > \\]
> > However, it is not true that $E(1/x) = 1/E(x)$. Then, I wonder how to ensure the inequality, when the gradient estimation involves randomness.

---

> ### Author Response · Authors · 2023-11-22
>
> Thank you for the follow-up question, and we are glad we clarified the other two questions.
>
> We are still a bit confused, as *we simply do not use the inequality*  $E(\eta_{t, k}^i) \leq E ( \frac{ \gamma ||  x_{t, k}^i - x_{t, k-1}^i  ||  }{ 2 || \nabla f_i( x_{t, k}^i ) -\nabla f_i (  x_{t, k-1}^i )  ||  }   ) $ anywhere in the proof.
> We agree that $E(1/x) \neq 1/E(x)$, but again, we do not use such equality nor the inequality above in our proof.

---

> > ### Comment · Reviewer_WWBM · 2023-11-23
> >
> > The major inequality is $||\nabla f(x_{t-1}) - \nabla f(x_t)\| \leq \frac{1}{2\eta_t} ||x_t - x_{t-1}||$, which is easyly to extend to stochastic version. I have no more concerns and raised my score.

---

> > > ### Author Response · Authors · 2023-11-23
> > >
> > > We truly appreciate your feedback and engagement throughout the discussion! Thank you again for taking the time to carefully review our paper and provide constructive feedback.

---

### Official Review · Reviewer_WDyG · 2023-11-03

**Soundness:** 3 good
**Presentation:** 3 good
**Contribution:** 3 good
**Rating:** 6
**Confidence:** 2

**Summary:**

Federated learning tasks are sensitive to the learning rate selection in client level. The paper argues it may be beneficial if we could enable different learning rate or learning rate scheduler per client due to the highly heterogeneous nature of FL clients. The paper proposes an auto-tuned learning rate scheduler, that could enable learning rate adapting to each client. The paper theoretically show the convergence of the proposed approach and experimental results demonstrate the effectiveness of $\Delta$-SGD.

**Strengths:**

Disclaimer: the reviewer is not very familiar with the hyperparameter auto-tuning in centralized computing or in distributed federated computing. Thus, I may not fairly assess the novelty of the technique proposed by this paper.

The paper is well written and easy to follow. I find the paper enjoyable to read.

The hyperparameter tuning in client level is tedious and existing literature typically by default set universal learning rate for each client. Therefore, the paper tackles an under-explored problem and proposes a simple and effective approach.

**Weaknesses:**

- Novelty and technical challenge: Client-level optimization is orthogonal to server-level optimization. Therefore, using auto-tuning, which has been studied extensively in the context of centralized computing, in client level is not a very challenging transferral, as we could directly use any auto-tuner directly in client level and check the performance of FL tasks.

I am wondering whether the auto-tuner, used or partially inspired by any practice in centralized computing. And is there any unique challenge if we simply combine any centralized auto-tuner to FL clients?

- Insufficient comparison with auto-tuner baseline: As I mentioned in the last point, it should not be very challenging to directly deploy auto-tuners to FL clients. However, there is limited comparison to this important baseline, i.e., some representative autotuners developed in centralized computing directly used in FL clients.

- Non-standard Assumptions used in theory: bounded gradient is strong assumption, and a bit contradictory to the heterogeneous setting the paper is motivated from. But it may be understandable to use bounded gradient as many existing literature especially who study adaptive optimizers also use it. Strong growth of dissimilarity is less standard and few papers in FL use it. Not sure whether it has any realistic and intuitive correspondence in FL tasks.

**Questions:**

My main questions lie in the previous weaknesses section.

Also an optional question: though it may be true there is no convergence guarantee given to the varying step size across clients, there are various papers that give convergence guarantee to the scenario where clients can have different number of local iterations, which seems to be a bit related. Is there any connection or difference in proving these two scenarios?

---

> ### Author Response · Authors · 2023-11-20
> **Thank you for the review!**
>
> Dear reviewer WDyG,
>
> We thank you for your review and comments. We are encouraged you found our paper well-written and enjoyable to read. As you noted, our work proposes a simple and effective approach that tackles an under-explored problem of client step size optimization in the federated learning setting. We hope our comments below clarify some of your concerns.
>
> ---
>
>
> > **Q: Applying auto-tuner to client level is simple?**
>
> **A:** We are confused about what you mean by “auto-tuner.” To clarify and help the discussion, assuming by auto-tuning you mean *adaptive methods*, Adam and Adagrad (popular adaptive methods) perform poorly (c.f., Table 1). SPS, a recently proposed adaptive method, also performs quite suboptimally in most cases. We kindly remind the reviewer of two works we cite [1, 2] that study the inefficacy of naively employing adaptive methods like Adagrad to the client optimizer, leading to suboptimal performance (similarly to what we show in Table 1 of our manuscript), and hence required server-side modification.
>
> The key difference is that our proposed method, *$\Delta$-SGD, performs well across all the settings without additional tuning or modification to the server-side aggregation.*
>
> ---
>
> > **Q: Non-standard assumptions like bounded gradient and strong growth of dissimilarity?**
>
> **A:** As the reviewer acknowledged, bounded gradient is quite a common assumption in nonconvex optimization [3, 4, 5, 6], especially when using an adaptive optimizer. We agree that strong growth of dissimilarity is non-standard, yet $\Delta$-SGD enables each client to use its own step size, which is a significant difference and advantage compared to other FL methods. Therefore, the local iterates can diverge from each other more rapidly (by using a larger step size), which is intuitively why we need a slightly non-standard assumption. We note that relaxing the assumptions can be an interesting future work.
>
> ---
>
>
> > **Q: Any connection to FL papers proving convergence where clients can have different numbers of local iterations?**
>
> **A:** We thank the reviewer for this comment. We do not see a clear connection between our proposed auto-tuned step size selection and existing analyses based on the number of local iterations. *A major difference is that using a different number of local iterations implies computing more gradients, whereas $\Delta$-SGD enables a simple step size calculation without additional computation.*
> Hyperparameters in gradient-based routines might indeed be connected. E.g., some works connect step size selection in stochastic cases with the mini-batch size per gradient calculation. We consider this question an interesting open question to study in the near future, but as of now, it is a bit orthogonal to this submission.
>
> ---
>
>
> We hope the above answers clarify your concerns, and if so, we hope you could consider reflecting on the score. Thank you again for your time in reviewing our manuscript.
>
>
> **References**
>
> [1] Wang, et al. (2021) “Local Adaptivity in Federated Learning: Convergence and Consistency”
>
> [2] Xie, et al. (2020) “Local AdaAlter: Communication-Efficient Stochastic Gradient Descent with Adaptive Learning Rates”
>
> [3] Ward, et al. (2019)  “AdaGrad Stepsizes: Sharp Convergence Over Nonconvex Landscapes”
>
> [4] Koloskova, et al. (2022) "Sharper convergence guarantees for asynchronous sgd for distributed and federated learning." Advances in Neural Information Processing Systems 35
>
> [5] Fallah, et al. (2020) "Personalized federated learning with theoretical guarantees: A model-agnostic meta-learning approach." Advances in Neural Information Processing Systems 33
>
> [6] Nguyen, John, et al. (2022) "Federated learning with buffered asynchronous aggregation." International Conference on Artificial Intelligence and Statistics.

---

> ### Comment · Reviewer_WDyG · 2023-11-23
> **Thanks for the rebuttal**
>
> Thanks authors for the detailed rebuttal. I am satisfied with the answers w.r.t. assumption and unequal number of local computation.
>
> With respect to the first point, i.e., auto-tuner part, I am not specifically referring to adaptive approaches like adam or adagrad. For example, there are many learning rate schedulers in centralized computing, e.g. warmup, multistage [3], cosine annealing [1], cyclic [2] and so on. And also some auto-tuning rule, e.g. yellowfin [4].
>
> These approaches should be easily used in client-level and could also alleviate some learning rate tuning efforts. I am wondering whether the authors could compare or at least discuss these works used in client level.
>
> [1] SGDR: Stochastic Gradient Descent with Restarts.
>
> [2] Cyclical Learning Rates for Training Neural Networks
>
> [3] A Stagewise Hyperparameter Scheduler to Improve Generalization
>
> [4] YellowFin and the Art of Momentum Tuning

---

> ### Author Response · Authors · 2023-11-23
>
> We're glad we clarified the other two questions.
>
> Thank you for clarifying what you meant by auto-tuners, and thanks for the references.
>
> We would like to point out that all the methods from the references either require users to set/tune multiple hyperparameters or solve multiple subroutines per iteration, as we detail below.
> *Importantly, what we describe below should be done for each client separately, in order for each client to use their own step size, as $\Delta$-SGD does.*
>
> ---
>
> > [1] SGDR: Stochastic Gradient Descent with Restarts
>
> This method requires the user to set $\eta_{min}^i, \eta_{max}^i, T_0,$ and $T_{mult}$, for the $i$-th run (Eq (5), pg. 4).
>
> > [2] Cyclical Learning Rates for Training Neural Networks
>
> This method similarly requires the user to set lower (base_lr) and upper (max_lr) bounds on the learning rate, as well as the frequency to cycle between the lower and the upper bounds (Figure 2, pg. 2)
>
> > [3] A Stagewise Hyperparameter Scheduler to Improve Generalization
>
> This method requires the user to set the "Quasi-Hyperbolic Momentum triplets" {$\alpha_i, \beta_i, v_i$} from $i=1$ to $M$, as well as the training length {$T_i$} from $i=1$ to $M$ (Algorithm 1, pg. 1532), where $M$ is the number of stage (another hyperparameter you have to set.)
>
> > [4] YellowFin and the Art of Momentum Tuning
>
> This method requires computing four different subroutines per iteration: (i) CURVATURE_RANGE, (ii) VARIANCE, (iii) DISTANCE, and (iv) SINGLE_STEP, as can be seen in Algorithm 1 (pg. 8).
>
> ---
>
> We do not have enough time to perform additional experiments to compare to these methods within the discussion period, but we will happily include these references in the related work, and try to add comparison results for the final version.

---

> > ### Comment · Reviewer_WDyG · 2023-11-23
> > **Thanks for your response**
> >
> > Thanks for the authors' response. My concern on this part is resolved.
> >
> > It seems the mentioned approaches may require several new hyperparameters to tune. But it still makes sense to discuss these approaches or compare with these approaches (maybe simply some default hyperparameters without carefully tuning to see what happens), as they represent efforts to auto determine learning rate in centralized setting and could be deployed to clients in a rather straightforward manner.
> >
> > I keep my recommendation of acceptance. I will discuss with other reviewers in the discussion period to consider increasing score.

---

### Author Response · Authors · 2023-11-22
**Response to all reviewers and AC**

We thank all the reviewers for their thoughtful and constructive feedback! Our paper proposes $\Delta$-SGD, a simple step size rule for SGD that enables each client to use their own step size in FL settings. We emphasize that $
\Delta$-SGD performs well *without any additional tuning* across *many different FL scenarios*. Simply put, $\Delta$-SGD achieves TOP-1 test accuracy in 73%, and TOP-2 accuracy in 100% of the experiments in Table 1.

Inspired by the reviewers' comments, we have additionally updated the supplementary material. Below, we summarize some of the major changes:
- In Appendix A.1, we clarified the proof of Lemma 2 to increase the readability.
- In Appendix B.1, we added additional experiments using FedAdam, a server-side adaptive method, to show that $\Delta$-SGD can be seamlessly combined with server-side adaptive methods.
- In Appendix B.2, we added additional experiments using the FedProx loss function, completing Table 2b of the original submission over all datasets.
- In Appendix B.3, we performed three independent trials and reported the averages and the standard deviations for each client optimizer for two tasks: CIFAR-10 classification trained with a ResNet-18, and FMNIST classification trained with a shallow CNN. $\Delta$-SGD not only achieves the best average test accuracy in both cases, but also enjoys extremely low standard deviation.
- In Appendix B.4, we visualize the step size conditions for $\Delta$-SGD, to show in practice how $\Delta$-SGD adapts its step size by approximating the local smoothness.

We hope the reviewers consider reflecting their scores considering these updates.
We once again thank the reviewers and the AC for their time in reviewing our manuscript.

---

### Meta-Review · Area_Chair_mAZp · 2023-12-08

**Metareview:**

This paper presents a novel method for automatically adjusting the local learning rate per-client in federated learning. Theoretical convergence guarantees are provided, and the empirical results on smaller datasets commonly used in the FL literature illustrate the promise of the approach to simplify federated training.

All reviewers agreed that the paper addresses an important problem, with clear motivation, that the results are clearly presented, the theory is sound, and the empirical performance is promising. The simplicity of the approach is a major asset, especially for practical applications of federated learning.

The paper could be strengthened by relaxing some of the strong assumptions, such as bounded gradients. It could also be strengthened by including comparisons to additional relevant methods, such as FedEx or by more clearly connecting this approach to work such as FedNova where clients take different numbers of steps / different learning local learning rates, while accounting for how this may impact fairness or introduce objective inconsistency.

On the whole, we recommend accept despite this room for improvement because the paper opens up a promising direction and addresses an important problem in federated learning.

**Justification For Why Not Higher Score:**

I would have recommended a higher score if:
* There was more substantial novelty (e.g., similar theoretical results but with weaker assumptions)
* Or if the empirical results were more substantially convincing, including results on larger-scale datasets
* Of if a more thorough comparison had been made with other baselines, such as FedEx

**Justification For Why Not Lower Score:**

The paper pinpoints an important problem and explores a novel approach which has not previously been studied for federated learning. This paper is likely to stimulate more discussion and work in the FL research community which will lead to progress in the field.

---

### Decision · Program_Chairs · 2024-01-16

Accept (poster)